# Notches and Fatigue on Aircraft-Grade Aluminium Alloys

**DOI:** 10.3390/ma17184639

**Published:** 2024-09-21

**Authors:** Valentin Zichil, Cosmin Constantin Grigoras, Vlad Andrei Ciubotariu

**Affiliations:** 1Department of Engineering and Management, Mechatronics, “Vasile Alecsandri” University of Bacău, 157 Calea Mărăsesti, 600115 Bacau, Romania; valentinz@ub.ro; 2Department of Industrial Systems Engineering and Management, “Vasile Alecsandri” University of Bacău, 157 Calea Mărăsesti, 600115 Bacau, Romania; vlad.ciubotariu@ub.ro

**Keywords:** aircraft-grade aluminium alloys, induced notches, low-cycle fatigue, statistical analysis

## Abstract

The influence of notches and fatigue on the ultimate tensile strength and elongation at break of aluminium alloys (2024-T3, 6061-T4, 6061-T4 uncoated, 6061-T6 uncoated, 7075-T0, and 7076-T6) is presented in this study. A total of 120 specimens were used. On all specimens, notches were made using a CNC machine, with 60 of them subjected to low-cycle fatigue (LCF) before undergoing the tensile test. Based on the statistical examination of the measured data, mathematical prediction models have been established. Compared to their unscratched counterparts, the results indicate a significant decrease in the UTS and elongation at break for both notched and notched-fatigued specimens. The LCF pre-treatment contributes to the negative impacts of the notches, resulting in reduced values for the UTS and elongation at break, thus concluding that surface integrity is critical for maintaining the structural strength of aircraft components.

## 1. Introduction

Exploring the effects of different factors on aircraft materials is a significant area of study. Researchers are currently investigating how scratches, impacts, and cyclic loading affect the durability and fatigue life of various aluminium alloys commonly used in aircraft construction. Understanding this information is essential to accurately forecasting component lifespans and maintaining aircraft safety.

Gaining insights into the behaviour of aluminium alloys when subjected to intense deformation conditions is of the utmost importance in enhancing the longevity and efficiency of components across diverse industries. Through the integration of numerical simulations and experimental scratch tests, researchers can acquire valuable data on the underlying wear mechanisms occurring on the surface of these alloys. This knowledge can then be utilized to enhance material design and processing strategies, resulting in more efficient outcomes. Experts are currently exploring novel approaches to improve the maintenance and safety of aircraft.

The field of aircraft inspection is rapidly advancing thanks to technological advancements such as the “Kiropter” climbing robot and the “Air-Cobot” collaborative robot. These robots capture high-resolution images of aircraft surfaces, analysing them to identify defects and stains. Researchers are also working on automated dent scanning and flaw detection algorithms to improve efficiency and precision. To test how well the model worked, real pictures of aircraft surfaces were used. These photos were taken by the remote-controlled robot of both a large Boeing 737 and a smaller plane [1]. 

In another study, the authors present a particle filter (PF)-based prediction algorithm that can predict crack propagation in aircraft construction using historical non-destructive testing data from an Airbus A310 aircraft. This method improves the precision of the proposed estimation, but existing simulation methods lack the necessary precision [2]. An automated vision-based inspection system was developed for identifying and analysing defects on aircraft surfaces, using local surface properties to identify undesired defects like dents, protrusions, or scratches. The Moving Least Squares (MLS) algorithm reduces noise and improves data quality. The system, which was tested on real aircraft data from an Airbus A320, has the potential for practical use in various industries [3]. 

In 2022, Avdelidis et al. presented a novel method for automating the recognition and classification of defects from visual images using an unmanned aerial vehicle (UAV) equipped with an image sensor. In the initial phase, the DenseNet201 CNN architecture achieved an impressive accuracy rate of 81.82%, and the proposed approach can achieve a perfect accuracy rate of 100% for the classification of missing or damaged exterior paint and primer, as well as dents [4]. 

In 2020, Xia et al. published their work, in which they present a 3D defect detection method for monitoring riveting and seam quality in high-value aircraft using fringe projection, automated identification, localisation, and augmented reality visualization. Current methods include rivet detection algorithms, non-destructive testing devices, hand-held systems, and innovative systems like structured light scanning and industrial close-range photogrammetry [5]. Maintenance, repair, and overhaul (MRO) companies are increasingly interested in the automation of aircraft inspections, which pose significant risks, require significant time and manpower, and are prone to mistakes. Automation holds great promise for enhanced safety, increased productivity, minimised aircraft downtime, and consistent output. UAVs can utilize pattern recognition, a prominent inspection technology that offers benefits, like simplicity, compatibility with standard hardware, and effectiveness in detecting scratches and lightning damage. Experts have suggested thermal imaging as a potential method for detecting delamination and corrosion [6].

In 2022, Yang et al. conducted research on the impact of scratches on the fatigue of ZL114A aluminium, a widely used alloy in aerospace and machinery. Both scratched smooth specimens and scratched hole plates underwent examination during the fatigue testing. The results indicate that scratches with depths of 0.15, 0.25, and 0.35 mm had a significant impact on the average fatigue life of the specimens. These scratches led to a reduction of approximately 35%, 50%, and 61% in the average fatigue life when compared to unscratched specimens. A fatigue damage model was developed using continuum damage mechanics (CDM) to evaluate the fatigue performance of actual components with surface imperfections [7]. Due to their thermoplastic nature, high strength-to-weight ratio, and corrosion resistance, aluminium alloys play a crucial role as load-bearing components in aerospace aircraft. Often subjected to cyclic loading in service, these alloys require excellent fatigue and damage tolerance properties. For material safety and security, fatigue crack propagation is critical. Recent research has concentrated on the impact of course second-phase particles on fatigue properties. In the high-cyclic stress (HCS) regime, cracks start where the Al7Cu2Fe second-phase particles meet the matrix. Previous fatigue strength modelling and life prediction based on HCS fracture characteristics are typically expressed in terms of the crack source’s effective area. However, previous studies have primarily concentrated on the size, type, and location of the fatigue sources, leaving quantitative studies on the number of fatigue sources limited [8].

Another approach proposes a new multiaxial fatigue life prediction model for aircraft aluminium alloys, specifically for the thin-walled tubular and notched specimens. The model defines a new characteristic plane (subcritical plane) to describe the particularity of additional cyclic hardening under nonproportional loading conditions. A corresponding damage parameter is built on this subcritical plane, analysing the dynamic path of the stress spindle, material property, and loading environment. Multiaxial fatigue refers to the fatigue failure behaviour of materials under multiaxial loads, which can be either proportional or nonproportional [9]. In the aerospace industry, fatigue crack propagation poses a significant problem, prompting the use of machine learning algorithms to identify the fatigue crack growth (FCG) rate. It can affect the design of aircraft structures, leading to safety issues and financial losses. However, these models have some flaws, such as not considering non-linearities and other factors like load ratio and fracture toughness. By 2022, researchers have proposed new research methodologies that combine numerical techniques with machine learning algorithms, like Bui’s knowledge-based neural network (KBNN) and Bhattacharya’s extended finite element method (XFEM) [10].

Li et al. proposed a different approach in 2024, examining the post-impact fatigue damage of the aluminium–lithium alloy, the material used in the C919 aircraft fuselage. It uses a residual stress–strain field from a quasi-static simulation to detect initial impact damage, as well as a continuum damage mechanics approach to forecast fatigue life. The study highlights the importance of post-impact fatigue life studies for the proper maintenance of aircraft structures. The study also emphasizes the influence of impact energy, impact pit dimension, and insert shape on residual fatigue life, as well as the impact of U-shaped inserts on fatigue life, highlighting the need for more research on fatigue properties. The continuum damage mechanics method is used for structural fatigue analysis, with impact tests and numerical simulations [11]. 

One of the least researched topics is the investigation of the vibration fatigue behaviour of aircraft aluminium alloys. In 2022, Teng et al. researched aluminium alloy 7050, a material that holds significant importance in the aerospace industry. Researchers established the experimental criteria for concluding the vibration fatigue experiment based on the reduction in acceleration and frequency values. Researchers have put forth different approaches to investigate the vibration fatigue behaviour of structures, such as conducting fatigue tests using MZGS-100PL and NI CRIO-9074 controllers [12].

In 2018, Rayno et al. developed a 3D scanning method to accurately assess surface damage on aircraft structural panels. The method showed 95% accuracy in dent depth measurements compared to a Starrett 643J dial depth gauge in 54 flat panel dents and 95% in 74 curved panel dents. Dent depth quantification involved comparing a point cloud rendering of the damaged surface with a surface fit that approximated the original, undamaged surface. By using convergence studies the efficiency and reliability of the technique were evaluated. Image processing techniques were used to measure dent length and area, demonstrating their potential for prompt extraction of surface dent measurements during on-site aircraft inspections [13]. 

Garcia et al. investigated crack propagation analysis using linear elastic fracture mechanics, considering computer simulations. The study focuses on cruciform specimens subjected to biaxial fatigue stresses and an initial 45°-angled crack, with a loading phase angle of 90° and 180°. This study was conducted using the extended finite element technique (XFEM) to investigate crack propagation, phase angles, and orientation criteria under nonproportional loading conditions, centred around engineering structures found in the automotive and aerospace industries, which frequently encounter multiaxial stresses that can lead to structural failures. Crack propagation can manifest either in a co-planar fashion or at a specific deviation angle, depending on the material’s characteristics and the type of stress applied. It focuses on analysing and comparing different nonproportional loading criteria, while also assessing the influence of displacement ratio and phase angle on crack orientation [14]. In a similar approach, the authors, by using computer simulation, examined fracture courses in three-wing spar designs and their impact on fatigue life calculation. It compared the efficiency of differential wing spars to two optimised integral spars using the extended finite element method (XFEM) and Morfeo/Crack software. The results showed significant strain-rate dependence for AA1050, while it was negligible for AA7075. A visco-plastic constitutive model based on Johnson–Cook was used to incorporate strain behaviour into the numerical simulations during high-velocity scratching [15].

Varga et al. developed a simulation framework in 2021 that targets meshless methods, which have the inherent capability to handle significant deformations and material removal that often arise in various processes. The Material Point Method (MPM) is used to accurately calculate strain rates and stresses. The study presents a potential work scheme to address the challenge of obtaining near-surface material properties at high strain rates using experimental tools and contributes to the development of a reliable and effective methodology in this field. The results indicate a significant strain-rate dependence for AA1050, while it is negligible for AA7075 [16]. 

In this experimental study, the defects are V-shaped with a considerable depth relative to the thickness of the specimens. Therefore, the term “notches” refers to these defects. Considering the research conducted in this field, the authors considered it necessary to conduct this study using six different types of aircraft-grade aluminium alloys, with notches orientated in different directions and having different lengths and depths. The study was conducted with a focus on the previously analysed scientific documentation, emphasising the importance of maintaining integrity and predicting the fatigue phenomenon, especially in the case of fuselage parts. The results were examined for relevance, performing a statistical analysis using the response surface methodology (RMS) approach and ANOVA for significance.

## 2. Aluminium Alloys and Methodology

This experimental study’s methodology includes testing 120 specimens of six aluminium aircraft-grade alloys with a thickness between 1 and 2 mm: 2024-T3 (Kaiser Aluminum, Lake Forest, CA, USA), 6061-T4 (AMAG, Braunau-Ranshofen, Austria), 6061-T4 uncoated (AMAG, Braunau-Ranshofen, Austria), 6061-T6 uncoated (Kaiser Aluminum, Lake Forest, CA, USA), 7075-T0 (Aleris Rolled Products Germany GmbH, Koblenz, Germany), 7075-T6 (Kaiser Aluminum, Lake Forest, CA, USA). Prior to the tensile test, the specimens were notched at different angles, lengths, and depths; 60 scratched specimens underwent a fatigue stress procedure. To evaluate the collected data, an ANOVA analysis was performed, focussing on the ultimate tensile strength (UTS) and elongation at break. The obtained data provides important insights into the material’s behaviour. These data help predict the service life and reliability of aluminium components exposed to intricate loading situations.

### 2.1. Aircraft-Grad Aluminium Alloys

A variety of technical fields extensively use aluminium alloys due to their superior corrosion resistance and attractive weight-to-strength ratio. Because of their unique mechanical characteristics and application-specific applicability, the alloys in the 2024, 6061, and 7075 series have attracted a lot of interest among the many available alloys.

The high-strength aluminium alloy 2024-T3 is well-known for having an exceptional strength-to-weight ratio. Through artificial age and solution heat treatment, the T3 temper imparts exceptional strength. Because of its exceptional mechanical qualities, it is a top choice for structural parts used in the aerospace sector, including aircraft landing gear, wings, and fuselages.

Two alloys that provide an excellent blend of strength, ductility, and weldability are 6061-T4 and 6061-T6. The T4 temper achieves moderate strength through natural ageing and solution heat treatment. On the other hand, the T6 temper experiences an extra artificial ageing process that increases its hardness and strength. Many different sectors, including maritime, construction, and transportation, find applications for both tempers. The mechanical qualities of uncoated 6061-T4 and 6061-T6 are comparable to those of coated; however, their corrosion resistance and surface features may differ.

Of the alloys under consideration, 7075-T0 has a higher UTS in its annealed (O) form. It is vulnerable to stress, corrosion, and cracking, however. The 7075-T6 temper uses artificial ageing and solution heat treatment to greatly increase strength and hardness. High-stress applications, like sports goods, military gear, and aircraft components, where weight reduction is essential, mostly employ this alloy.

Tensile testing on an INSTRON 8801 servohydraulic fatigue testing system (INSTRON, Norwood, MA, USA) with a 100 kN loading cell and a 2630-107 axial clip-on extensometer (INSTRON, Norwood, MA, USA) determined the mechanical properties given in Table 1. For each alloy, five specimens were used orienting the load in the rolling direction. For each set of results, the arithmetic mean of the obtained values was considered.

The specimens were cut into shape using an Oreelaser PH3015 CNC laser machine (Oreelaser, Jinan, China), following the specific configuration shown in Figure 1. At this stage, ensuring the accuracy of the 12.5 mm part’s width (W) was important because it directly affected the calculation of the cross-sectional area (CSA). This measurement is critical for determining internal stress; therefore, it was measured before testing, with a tolerance of ±0.1 mm.

### 2.2. Experimental Plan

The experimental plan was designed using the DesignExpert software (Stat-Ease, Minneapolis, MN, USA, 2024, v. 12.0.3). This study used a Response Surface design with a randomized subtype and a D-optimal design, suggesting 60 specimens. Alloys were classified as nominal categorical components, and numerical discrete factors were also assigned to the plan, as indicated in Table 2.

Initially, an experimental strategy was devised that considered the six alloys, their varying thicknesses, and the direction, length, and depth of the notch. Due to the integration of a significant number of specimens, this experimental design proved to be advantageous in facilitating the systematic and efficient execution of the notches. After the notch execution, the experimental plan was adapted to utilize the cross-sectional area (CSA). As the applied force is reported to the cross-section area (Figure 2), the increase in stress will occur at the point where the specimen is notched, as in this section, there is a smaller cross-sectional area. Therefore, the final form in which the data was analysed is that of Table 2, which considers the 6 types of aircraft-grad aluminium alloys (2024-T3, 6061-T4, 6061-T4 uncoated, 6061-T6 uncoated, 7075-T0, 7075-T6), 6 material thicknesses (1.0, 1.2, 1.27, 1.6, 1.8, 2.0 mm), 3 notch orientation directions (0°- in the rolling direction, 45°- to the rolling direction, 90°- perpendicular to the rolling direction; the applied load direction, during the tensile test, was aligned to the rolling direction), and resulting cross-sectional area (calculated for each specimen). As for the responses, 4 sets of data were analysed regarding the ultimate tensile strength and elongation at break of the notched (UTS_n_, ε_n_) and the notched-fatigued specimens (UTS_nf_, ε_nf_).

### 2.3. Notch Methodology

To obtain accurate control over the previously mentioned indicators, all the scratches were made using a Knuth Rapimill 700 CNC milling machine (Knuth, Wasbek, Germany) and a tool with a tip angle of 60° (Figure 3a,b). Based on the specified dimensions of the notch and to ensure precise execution and reputability, a dedicated program was written in CNC machine code, setting a rotation speed of 1500 rpm and a feed of 5 mm/s. The tool used for executing the scratches was manufactured on a lathe, resulting in a maximum measured radial runout of 0.07 mm. The tip angle of 60° was measured using a Mitutoyo PH-3515 profile measuring projector (Mitutoyo, Kawasaki, Japan).

The centre point of each notch is in the centre of each specimen; at this point, the scratches could be executed as previously described:3 orientation directions (Figure 4a): along the path of force application (0°), perpendicular to this direction (90°), and at an intermediate angle (45°).3 notch lengths, as a percentage of the available length, based on the angle of orientation, are as follows (Figure 4b): the whole length (100%), half of the length (50%), and one-fourth of the length (25%).3 notch depths (Figure 4c), which are expressed as percentages of the specimen thickness: 50%, 25%, and 12.5%.Considering the 3 orientations with 3 lengths (Figure 4d–f) and 3 depths, a total of 27 different combinations were executed.

The methodology used for the execution of the notches required validation to ensure control over the resulting geometry. The process precision and the wear of the scratching tool were assets by cutting a 5 mm-thick slice from every 10th specimen using a Micromet Evolution metallographic cutting machine (Remet, Bolognia, Italy). Linear and angular measurements were made on the Mitutoyo PH-3515 profile measuring projector. No significant differences were identified between the measurements and the programmed parameters (depth and length); therefore, Equations (1)–(5) were used to generate the dimensions indicated in Table 3. The cross-section area (CSA) was determined based on the direction of the force exerted during the tensile test, specifically a plane that is perpendicular to the force’s direction and, in normal conditions, has a cross-section area as indicated in Figure 5a (CSA = W × T).

Examining this plane, as shown in Figure 5b–d, requires subtracting the area of the notch, which varies depending on the direction of the notch, its width (NW), its depth (ND), and its length (NL). Equations (1)–(5) present the mathematical formulas used to calculate the remaining cross-section area.
CSA_SS_ = W × T,(1)
CSA_0°notch_ = CSA_SS_ − ND^2^ × tan(30°),(2)
CSA_45°notch_ = CSA_SS_ − SW^2^ × ND × sin(45°), where NW = NW’ × sin(45°),(3)
CSA_90°notch_ = CSA_SS_ − [ND^2^ × tan(30°) + (ND × NL)] − used for 12.5% and 50% of the notch length,(4)
CSA_90°notch_ = W × (T − ND) − used for 100% of the notch length.(5)

Therefore, based on the previously presented methodology, the lengths and depths of the scratches, as well as the cross-section area, were calculated for each alloy, considering the notch directions and the thickness of each specimen. The complete data are available in Table 3.

**Table 3 materials-17-04639-t003:** Notch orientations, dimensions, and cross-section area for each alloy.

Aluminium Alloy	Notch Direction °	Specimen Thickness mm	Notch Length in mm as 100, 50, and 25% of Maximum Available Length		Notch Depth in mm as 50, 25, 12.5% of Specimen Thickness		Cross-Section Area mm^2^
2024-T3	0	1.0	25%	18.75	12.5%	0.125	12.49
			50%	37.5	50%	0.5	12.36
		1.6	50%	37.5	12.5%	0.2	19.98
			100%	75.0	12.5%	0.2	19.98
		1.8	100%	75.0	50%	0.9	22.03
		2.0	100%	75.0	12.5%	0.25	24.96
	45	1.0	100%	17.678	12.5%	0.125	12.49
		1.2	100%	17.678	25%	0.3	14.93
		2.0	25%	4.419	50%	1.0	24.18
			100%	17.678	50%	1.0	24.18
	90	1.2	12.5	12.5	12.5%	0.15	13.13
			3.125	3.125	25%	0.30	14.01
		1.6	6.25	6.25	25%	0.40	17.41
		1.8	3.125	3.125	12.5%	0.225	21.77
			12.5	12.5	25%	0.45	16.88
6061-T4	0	1.27	25%	18.75	12.5%	0.159	15.86
			100%	75.0	12.5%	0.159	15.86
			100%	75.0	50%	0.635	15.64
		1.6	25%	18.75	50%	0.8	19.63
	45	1.27	12.5%	4.419	50%	0.635	15.55
		1.6	12.5%	4.419	12.5%	0.2	19.97
			50%	8.839	25%	0.4	19.87
			100%	17.678	50%	0.8	19.48
	90	1.27	100%	12.5	50%	0.635	7.94
		1.6	25%	3.125	25%	0.4	18.66
6061-T4 uncoated	0	2.0	25%	18.75	12.5%	0.25	24.96
	45	2.0	100%	17.678	50%	1.0	24.18
	90	2.0	50%	6.25	12.5%	0.25	23.40
		2.0	50%	6.25	25%	0.5	21.73
		2.0	100%	12.5	50%	1.0	12.50
6061-T6 uncoated	0	1.6	50%	37.5	12.5%	0.2	19.98
		1.6	50%	37.5	50%	0.8	19.63
		1.6	100%	75.0	12.5%	0.2	19.98
	45	1.6	100%	17.678	50%	0.8	19.48
	90	1.6	100%	12.5	50%	0.8	10.00
7075-T0	0	1.6	25%	18.75	12.5%	0.2	19.98
		1.6	50%	37.5	25%	0.4	19.91
	45	1.0	100%	17.678	25%	0.25	12.45
		1.6	100%	17.678	12.5%	0.2	19.97
			12.5%	4.419	25%	0.4	19.87
	90	1.0	50%	6.25	12.5%	0.125	11.71
		1.0	100%	12.5	25%	0.25	9.38
		1.0	25%	3.125	50%	0.5	10.79
		1.0	100%	12.5	50%	0.5	6.25
		1.6	25%	3.125	50%	0.8	17.13
7075-T6	0	1.27	100%	75.0	50%	0.635	15.64
		1.8	25%	18.75	25%	0.45	22.38
		2.0	25%	18.75	12.5%	0.25	24.95
	45	1.0	50%	8.839	12.5%	0.125	12.49
			100%	17.678	12.5%	0.125	12.49
		1.27	25%	4.419	12.5%	0.159	15.85
		1.6	25%	4.419	12.5%	0.2	19.97
		1.8	25%	4.419	25%	0.45	22.34
			100%	17.678	25%	0.45	22.34
		2.0	25%	4.419	50%	1.0	24.18
	90	1.0	100%	12.5	12.5%	0.125	10.94
		1.27	100%	12.5	12.5%	4.419	13.89
		1.6	25%	3.125	50%	0.8	17.13
			100%	12.5	50%	0.8	10.00
		2.0	100%	12.5	12.5%	0.25	21.88

### 2.4. Fatigue Setup and Tensile Test

The fatigue and tensile tests were conducted using an Instron 8801 servohydraulic fatigue testing system with a 100 kN loading cell and a 2630-107 axial clip-on extensometer. WaveMatrix3 Dynamic Testing Software (INSTRON, Norwood, MA, USA, 2023, version 3) was used for the low-cycle fatigue (LCF), and Bluehill Universal Software (INSTRON, Norwood, MA, USA, 2023, version 4.42) was used for axial static testing.

A total of 60 specimens were subjected to fatigue tests through bending. The fatigue setup consists of a fixed and mobile component (Figure 6a). An S500 steel fixture serves as the fixed component, while two PLA 3D-printed 75 mm-radius dome-shaped geometries serve as specimen supports (Figure 6b). The upper part has four threaded holes on each corner, while M4 screws go through the lower part, fixing it around the specimen. For the mobile component of the setup, a custom modular fixture support was manufactured out of S500 steel alloy. The gripping jaws are knurled and tightened at a torque of 15 Nm using a Unior 266 dynamometric wrench (Unior, Zreče, Slovenia).

Testing different combinations of process parameters led to breaking the specimens with 50% notch depth. At this stage, the goal was to test the notched parts without breaking them. Therefore, as shown in Figure 6b,c, a symmetrical alternating cycle was used with a maximum amplitude of 6 mm and a 1 Hz cycle for 200 cycles.

All 120 specimens were subjected to the tensile test, conducted at a strain rate of 10 mm/min, with the load direction perpendicular to the transversal plane, as indicated in Figure 7a. This implies that the load is in the direction of the 0° notches and perpendicular to those oriented at 90°. The cross-section area for each specimen is indicated in Table 3 The tensile test ended when the specimen broke (Figure 7b), with the recorded data being processed for UTS and elongation at break.

## 3. Results and Statistical Analysis

Due to their favourable mechanical characteristics, aluminium alloys find widespread use in a range of technical applications. Defects, such as scratches, can, on the other hand, significantly undermine a structure’s structural integrity, necessitating a thorough understanding of their impact on mechanical properties. In the present study, the authors investigated how scratching and subsequent fatigue affect the tensile properties of aluminium alloys. The focus is on how the direction of the notch and the cross-sectional area impact the ultimate tensile strength (UTS) and elongation at break across multiple aluminium alloys (2xxx, 6xxx, and 7xxx series). The objective is to better understand the causes of mechanical property decrease by examining the failure processes of notched and notched-fatigued specimens.

The experimental data, which include the ultimate tensile strength (UTS) and elongation at break, were subjected to statistical analysis to detect patterns and relationships. The analysis uses regression analysis to create mathematical models that establish the relationship between the UTS, elongation at break, and factors such as notch direction and cross-sectional area. The model fit statistics were assessed to evaluate the predictive capabilities of the models.

Figure 8, which displays several of the specimens, reveals the failure mechanism. The image representation shows that the fracture originated and propagated from the notch.

For aluminium alloy 2024-T3, a total of 30 specimens were processed. The specimens are categorised into two groups: the notch-tensile test and the notch-fatigued-tensile test. Each group consists of 15 specimens. Within each group, there are six specimens with the notch orientated at 0°, four specimens at 45°, and five specimens at 90°. The results are shown in Table 4.

The data obtained for UTS and elongation at break were subjected to statistical analysis and separated into four distinct sets, as shown in Table 2. The resultant models are quadratic (the highest power of the independent variable is 2) or 2FI (two-fixed-one-random effect), which allow for the assessment of the impact of each element and the interaction between them on the outcome. The ANOVA analysis indicated that these models are statistically significant. The data presented in Table 5 indicate a *p*-value of less than 0.001 (the *p*-value is a statistical measure used to assess the significance of experimental results). The UTS_nf_ data indicated the need for an inverse square transformation, which was subsequently performed. The generated mathematical equations also emphasised this transformation.

Fit statistics play an important role in evaluating the degree to which a statistical model accurately matches a given collection of observable data. They help evaluate the model’s fidelity by reflecting the fundamental patterns within the given data. The values shown in Table 6 represent the goodness-of-fit metrics and indicate a significant level of confidence. R^2^ and the Adjusted R^2^ measure the model’s fit to the existing data, while the Predicted R^2^ analyses its ability to predict outcomes on fresh data. Adequate precision assesses the model’s dependability in predicting responses. Furthermore, the Model F-value and Lack of Fit F-value were calculated (the Model F-value indicates the presence of a correlation between the variables, while the Lack of Fit F-value tells if the chosen model form is correct).

UTS_n_—a Model F-value of 860.15 indicates that the model is statistically significant. The probability of an F-value of this magnitude occurring solely due to noise is just 0.01%; *p*-values below 0.0500 imply that the model terms are statistically significant. In this case, the relevant model terms are ND, CSA, material, and ND^2^. The F-value of 4.56 for the Lack of Fit indicates that the Lack of Fit is statistically significant. The probability of a Lack of Fit F-value of this magnitude occurring solely due to noise is just 3.21%. The Predicted R^2^ value of 0.9277 shows a strong correlation with the Adjusted R^2^ value of 0.9266, indicating a high level of agreement between the two, the difference between these values being less than 0.2.ε_n_—a Model F-value of 120.93 indicates that the model is statistically significant; the *p*-values are below 0.0500 and imply that the model terms are statistically significant (in this scenario, the model terms ND, CSA, material, and the interaction between material and ND have major importance). An F-value of 33.44 for the Lack of Fit indicates that the Lack of Fit is statistically significant. The Predicted R^2^ value of 0.9115 shows a strong correlation with the Adjusted R^2^ value of 0. 8934.UTS_nf_—a Model F-value of 175.70 indicates that the model is statistically significant; the *p*-values are below 0.0500 and indicate that the model terms are statistically significant, the relevant model terms being ND, CSA, material, and ND^2^. The F-value of 3.80 for the Lack of Fit indicates that the Lack of Fit is statistically significant. The probability of a Lack of Fit F-value of this magnitude occurring solely due to noise is just 4.98%. The Predicted R^2^ value of 0.8990 is quite consistent with the Adjusted R^2^ value of 0. 8734, indicating a discrepancy of less than 0.2.ε_nf_—with a Model F-value of 97.40 the model is statistically significant. The probability of an F-value of this magnitude, occurring solely due to noise, is just 0.01%; a *p*-value below 0.0500 implies that the model terms are statistically significant; ND, CSA, material, and the interaction between material and ND have major significance. A Lack of Fit F-value of 12.26 indicates that the Lack of Fit is statistically significant. The probability of a Lack of Fit F-value of this magnitude occurring solely due to noise is just 0.23%. The Predicted R^2^ value of 0.8804 shows a strong correlation with the Adjusted R^2^ value of 0. 8703.

### 3.1. UTS

#### 3.1.1. Aluminium Alloy 2024-T3

The average UTS_n_ of specimens notched at 0° is 444.683 MPa, whereas the UTS_nf_ of notched-fatigued specimens is 376.959 MPa, resulting in a difference of 17.966%. The percentage decreases to 29.854% for specimens notched at 45°, from an average of 424.351 to 326.791 MPa. Specimens with notches at 90° result in a significant loss in UTS, dropping from an average of 414.581 to 319.939 MPa, a decrease of 29.581%. Compared to the data from Table 1, the overall average UTS_n_ dropped by 5.172% and the UTS_nf_ by 31.876%.

The ANOVA analysis generated two mathematical models, represented by Equations (6) and (7), which correspond to the 2024-T3 alloy UTS_n_ and UTS_nf_. The estimate of UTS reveals both the individual significance of notch direction (NDir) and CSA, as well as their combined effect. The first case established a polynomial relationship, while Equation (7) underwent the inverse square transformation as mentioned in Table 4. The model from Equation (6) includes a quadratic relationship between UTS_n_2024-T3_, NDir, and CSA. This means that these parameters do not have a linear effect on the strength of the material. The negative coefficient for the NDir term indicates that increasing the notch direction generally decreases UTS. Conversely, the positive coefficient for the CSA term suggests that increasing cross-sectional area enhances UTS. The interaction term (NDir x CSA) and the quadratic terms for NDir and CSA further refine the model’s predictive capabilities. Equation (7), upon analysis, models the inverse of UTS_nf_2024-T3_ as a linear combination of NDir, CSA, and their interaction terms, indicating a distinct functional relationship from the first model. Similar to the previous model, the negative coefficient for the NDir term indicates that the notch direction has a detrimental effect on UTS. However, the positive coefficient for the CSA term implies a less pronounced cross-sectional area influence on UTS compared to the notched specimens. The inclusion of quadratic terms for NDir and CSA suggests a complex relationship.
UTS_n_2024-T3_ = 413.48289 − 0.431968 × NDir + 1.6023 × CSA − 0.00573 × NDir × CSA + 0.002552 × NDir^2^ + 0.002475 × CSA^2^,(6)
(7)1UTSnf2024−T3=0.057796+0.000106×NDir−0.000526×CSA−5.04659×10−7×NDir×CSA−5.68317×10−7×NDir2+9.98623×10−6×CSA2.

Figure 9a,b visually represents Equations (6) and (7), respectively; the graphs are constructed depending on the orientation of the notch and the remaining cross-sectional area following the notch. Green represents lower UTS values, whereas yellow represents higher values. The highlighted estimations are derived from both the input data range and extrapolated beyond it, using the predictions generated by these equations. Their comparison reveals that fatigue significantly influenced the results, outweighing the impact of simply notching the specimens. Therefore, instead of 280–380 MPa, the UTS falls within the range of 400 to 450 MPa. An inverse relationship was established between the UTS and both the notch angle and the cross-sectional area. As mentioned earlier, in Figure 2, the limit stress point is the point where the cross-sectional area is at its smallest, with the force acting perpendicular to the direction of the notch.

It is clear from comparing the two plots that fatigue significantly affects the mechanical properties of the aluminium alloy 2024-T3. The UTS values in Figure 9b are much lower. In addition, the cross-sectional effect plays a more significant role in the case of notched-fatigued specimens.

The statistics given emphasise the significance of considering both the direction of the notch and the cross-sectional area when assessing the mechanical properties of aluminium alloy 2024-T3, especially in the context of fatigue. The results suggest that these parameters are essential for understanding the material’s performance in applications that experience cyclic stress and surface deterioration.

#### 3.1.2. Aluminium Alloys 6061-T4, 6061-T4 and -T6 Uncoated

The UTS of the 6061-T4 alloy decreases as follows: from 255 MPa to an average of 227.822 MPa (11.929%) for the notched specimens and then further to 189.417 MPa (34.624%) for the fatigued specimens. Examining UTS_n_ and UTS_nf_ at various notch directions reveals a decrease of 21.291% for scratches made at 0°, 22.727% for 45°, and 16.568% for 90°.

Equations (8) and (9) highlight the mathematical models. Following prior indications from the 2024-T3 alloy, strong correlations were identified between the notch direction, cross-sectional area, and UTS. The models consider both the individual impact of each factor and their combined effect.
UTS_n_6061-T4_ = 211.41408 − 0.628555 × NDir + 2.35351 × CSA − 0.00573 × NDir × CSA + 0.002552 × NDir^2^ + 0.002475 × CSA^2^,(8)
(9)1UTSnf6061−T4=0.072142+0.000126×NDir−0.00033×CSA−5.04659×10−7×NDir×CSA−5.68317×10−7×NDir2+9.98623×10−6×CSA2.

In the case of notched specimens, the predictability of UTS_n_ values follows a linear pattern (Figure 10a), increasing as the CSA and NDir increase. However, for the UTS_nf_ (Figure 10b), increasing the notch angle beyond 67.5° and decreasing the CSA below 10 mm^2^ results in a sharp drop of 10 MPa in ultimate tensile strength.

The measurement results for the uncoated 6061-T4 alloy show an overall decrease of 6.576% in UTS_n_ and a reduction of 41.6% in UTS_nf_. It is important to note that this case represents the greatest percentage difference among all the alloys examined. By increasing the notch angle, the ultimate tensile strength decreases by 15 units, while the UTS_nf_ decreases by 12 units. For the specimens that underwent fatigue testing, the average ultimate tensile strength decreased by 32.886%, from 231.76 MPa to 174.432 MPa.

Previous observations indicate a consistent pattern in the mathematical models (Equations (10) and (11)), highlighting the correlation between the direction of the notch and the cross-sectional area. An important observation arises from the analysis of Figure 11b, constructed using Equation (11). It is noticeable that in contrast to the graph in Figure 10b, there are no sudden shifts. Instead, lower values were observed for the UTS_nf_ at 0° and 90°, with the highest values observed around 45°. Regarding the UTS_n_, as shown in Figure 11a, a minimum value is found at 0°, followed by increases up to 67.5°, and then a decrease up to 90°. The increase in the cross-sectional area results in a corresponding increase in UTS_n_ of up to 20 units.
UTS_n_6061-T4_uncoated_ = 211.00629 − 0.222059 × NDir + 1.13329 × CSA − 0.00573 × NDir × CSA + 0.002552 × NDir^2^ + 0.002475 × CSA^2^,(10)
(11)1UTSnf6061−T4uncoated=0.087737+0.000065×NDir−0.000783×CSA−5.04659×10−7×NDir×CSA−5.68317×10−7×NDir2+9.98623×10−6×CSA2.

The uncoated 6061-T6 alloy’s behaviour shows an average decrease of 298.061 MPa (7.244%) for the notched specimens and as low as 242.136 MPa (28.027%) for the notched-fatigued specimens. In this instance, the occurrence of fatigue after 200 cycles results in a decrease of 20.784% in UTS. Increasing the notch angle concerning the load direction causes the influence on the notched specimens to decrease, from 306.074 MPa to 293.669 MPa to 267.44 MPa. In this situation, fatigue leads to a decrease in values ranging from 237.872 to 247.839 MPa, with an average reduction of 19.355%.

Equations (12) and (13) display mathematical models similar to the previous ones, which Figure 12a,b graphically illustrates. Peak values of UTS_n_ fall between 45–67.5°, while scratches oriented at 0° exhibit lower values. The cross-sectional area is reduced from 22.5 to 7.5 mm^2^, resulting in a 40-unit drop in UTS_n_. Regarding the UTS_nf_, there is a noticeable pattern with respect to the 45°. It experiences a gradual decrease from 0° to 45°, with an increment of 10 units. Subsequently, there is a minor further increase of 10 units from 45° to 90°. Therefore, it can be deduced that the critical scratches are orientated at 45°. This, combined with a cross-sectional size of 5–6 mm^2^, results in the lowest UTS_nf_ values.
UTS_n_6061-T6_uncoated_ = 239.80017 − 0.235858 × NDir + 3.28308 × CSA − 0.00573 × NDir × CSA + 0.002552 × NDir^2^ + 0.002475 × CSA^2^,(12)
(13)1UTSnf6061−T6uncoated=0.065937+0.000064×NDir−0.000320×CSA−5.04659×10−7×NDir×CSA−5.68317×10−7×NDir2+9.98623×10−6×CSA2.

#### 3.1.3. Aluminium Alloys 7075-T0 and 7075-T6

It is shown that the UTS_nf_ for the 7075-T0 alloy drops by 23.445% (from 255.091 to 206.644 MPa) when notched at 0°, 23.646% (from 233.523 to 188.872 MPa) when notched at 45°, and 10.944% (from 215.3 to 194.063 MPa) when notched at 90°. In this case, fatigue has a significant impact, resulting in a reduction in UTS by as much as 21.49% when comparing the notched specimens to the notched-fatigued specimens. The results indicate a significant average decrease of 32.298% in comparison to the baseline UTS.

The ANOVA analysis revealed a significant correlation between NDir and CSA in terms of outcome. Equations (14) and (15) express the mathematical models derived in this instance, and Figure 13a,b show their graphical representation. The orientation of the notch in this instance is not optimal. It results in a maximum UTS_n_ value ranging from 200 to 230 MPa, corresponding to a CSA between 5 and 25 mm^2^. For UTS_nf_, there is a tendency for the values to decrease as NDir changes towards the end of the analysed interval.
UTS_n_7075-T0_ = 213.53762 − 0.401177 × NDir + 2.06010 × CSA − 0.00573 × NDir × CSA + 0.002552 × NDir^2^ + 0.002475 × CSA^2^,(14)
(15)1UTSnf7075−T0=0.075151+0.000062×NDir−0.000443×CSA−5.04659×10−7×NDir×CSA−5.68317×10−7×NDir2+9.98623×10−6×CSA2.

For notched specimens, the models from Equations (16) and (17) directly predict UTS, whereas the model for notched-fatigued specimens uses an inverse square relationship. This suggests fundamentally different relationships between the variables and the UTS under the two conditions.

Both models include quadratic terms for NDir and CSA, indicating non-linear relationships between the UTS (or 1/UTS) and these parameters. However, the specific coefficients and interactions between these terms likely differ, suggesting varying degrees of curvature in the relationships. In both models, the coefficients for the linear, interaction, and quadratic terms are different. This means that UTS is sensitive to notch direction and cross-sectional area in different ways. The differences between the models indicate that whether the specimen has undergone fatigue significantly influences the relationship between UTS, notch direction, and cross-sectional area. The inclusion of quadratic terms and the use of an inverse relationship in the fatigued case highlight the complexity of the degradation mechanisms under these conditions.
UTS_n_7075-T6_ = 496.1882 − 0.412119 × NDir + 1.48717 × CSA − 0.00573 × NDir × CSA + 0.002552 × NDir^2^ + 0.002475 × CSA^2^,(16)
(17)1UTSnf7075−T6=0.053912+0.000083×NDir−0.000513×CSA−5.04659×10−7×NDir×CSA−5.68317×10−7×NDir2+9.98623×10−6×CSA2.

The plot from Figure 14a of notched specimens indicates the correlation between ultimate tensile strength and the direction of the notch. As the notch angle increases, there is a consistent pattern of decreasing UTS. This implies that the direction of the notch has an important impact on the material’s strength. The influence of cross-sectional area on ultimate tensile strength (UTS) seems to be less significant since there is only a slight increase in UTS for higher cross-sectional areas. Figure 14b highlights the results for the notched-fatigued specimens, indicating an intricate correlation between the variables. While the effect of notch direction on ultimate tensile strength (UTS) remains noticeable, it appears to be less significant compared to the notched specimens. A noticeable correlation is observed between the cross-sectional area and the ultimate tensile strength (UTS), where an increase in the cross-sectional area leads to a significant decrease in UTS. The downward slope of the colour bands illustrates this relationship. The results suggest that fatigue significantly reduces the strength of the material, and the impact of the cross-sectional area becomes more important in determining the ultimate tensile strength (UTS) under such circumstances.

Upon comparing the two graphs, it is noticeable that fatigue has a substantial negative impact on the mechanical characteristics of the aluminium alloy 7075-T6, as the UTS values in Figure 14b are much lower. In addition, the impact of notch direction and cross-sectional area varies between the two factors, with the cross-sectional area having a more significant effect in the case of notched-fatigued specimens.

### 3.2. Elongation at Break

#### 3.2.1. Aluminium Alloy 2024-T3

The elongation at break of the notched (ε_n_) and notched-fatigued (ε_nf_) specimens decreased by 16.941% for the specimens notched at 0°, by 31.356% for those notched at 45°, and by 20.22% for those notched at 90°. The ε_n_ decreased by up to 152.155% compared to the values in Table 1, whereas the ε_nf_ decreased by 210.631%.

Both models from Equations (18) and (19) aim to predict the elongation at break of aluminium alloy 2024-T3 specimens by considering the notch direction and cross-sectional area. Nevertheless, they contain distinct variables and indicate diverse levels of complexity. Both models show linear effects for NDir and CSA, indicating that these factors have a direct impact on elongation at break. The models have an interaction factor (NDir × CSA) that considers how the direction of the notch and the cross-sectional area affect the length at break. The ε_nf_2024-T3_ model has a lower intercept value (6.64024) in contrast to the ε_n_2024-T3_ model (7.95718), indicating a typically reduced elongation at break for specimens that are notched-fatigued. The NDir coefficient for both models is negative, showing that an increase in notch direction often leads to a reduction in elongation at break. However, it is much more important in the ε_nf_2024-T3_ model, which means that the direction of the notch has a bigger impact on the lengthening of notched-fatigued specimens. Both models have a positive CSA coefficient, indicating that increasing the cross-sectional area typically leads to an increase in elongation at break. In the ε_nf_2024-T3_ model, the coefficient is slightly greater, indicating a stronger effect of cross-sectional area on elongation for notched-fatigued specimens. In both models, the interaction term has opposite signs. In the ε_n_2024-T3_ model, the result is negative, indicating a decreasing impact of the combined NDir and CSA on elongation. The ε_nf_2024-T3_ model has a positive correlation, indicating an increasing impact. The ε_nf_2024-T3_ model has quadratic factors for NDir and CSA, suggesting a non-linear correlation between these variables and elongation at break. This implies a more complex behaviour for notched-fatigued specimens. For aluminium alloy 2024-T3, the variations in the models indicate that fatigue has a major impact on the correlation between notch direction, cross-sectional area, and elongation at break.
ε_n_2024-T3_ = 7.95718 − 0.079094 × NDir + 0.096159 × CSA − 0.00015 × NDir × CSA,(18)
ε_nf_2024-T3_ = 6.64024 − 0.08116 × NDir + 0.099137 × CSA + 0.000227 × NDir × CSA + 0.00012 × NDir^2^ − 0.000647 × CSA^2^.(19)

Both plots use a colour gradient to display the elongation at break values, where warmer colours (red and orange) indicate greater elongations and cooler colours (blue and green) reflect lower elongations. The *x*-axis denotes the angle of the notch direction in degrees, while the *y*-axis shows the cross-sectional area of the notched specimen in square millimetres.

The contour plot from Figure 15a indicates a consistent pattern of decreasing elongation at break as the notch direction increases. This indicates that the direction of the notch has a major impact on the material’s ability to deform without breaking. Furthermore, as the cross-sectional area increases, this impact is less significant than the effects of the notch direction.

Figure 15b’s contour plot for notched-fatigued specimens shows a more complex pattern for the notched specimens. The impact of the direction of the notch on elongation remains apparent since lower notch angles are associated with larger elongations. Nevertheless, the impact of the cross-sectional area tends to be more prominent since the contour lines are more densely packed in the vertical direction. These results indicate that the simultaneous occurrence of a notch and fatigue greatly diminishes the material’s ability to stretch, and the size of the cross-sectional area becomes a more important factor when determining the extent of elongation at the fracture point.

Upon comparing the two graphs, it is noticeable that fatigue significantly reduces the elongation at break of the aluminium alloy 2024-T3. Moreover, the impact of cross-sectional area becomes significantly greater when fatigue is present, indicating that the material becomes more responsive to changes in shape under these circumstances.

#### 3.2.2. Aluminium Alloys 6061-T4, 6061-T4 and -T6 Uncoated

The analysis of the results for the 6061-T4 alloy indicates an average elongation at break of 6.084% for the notched specimens, compared to 5.148% for the notched-fatigued ones. The percentage difference between the notched and notched-fatigued specimens in the 0° and 90° notch directions ranges from 16.7 to 17%, with a significant difference of 20.793% for the 45° orientation. The initial material properties from Table 1 show a decrease in elongation results of 310.942% for the notched specimens and 385.641% for the notched-fatigued specimens.

In both Equations (20) and (21), there is an interaction term (NDir x CSA) that considers how the direction of the notch and the cross-sectional area affect the length at break. The ε_nf_6061-T4 model_ has a lower intercept value (5.48139) compared to the ε_n_6061-T4_ model (6.1297), suggesting a generally reduced elongation at break for specimens that are scraped and exhausted. Both models have a negative NDir coefficient, indicating that an increase in notch direction results in a decrease in elongation at break. However, the rate of elongation is almost the same in all models, suggesting that the influence of notch direction on elongation is comparable for both notched and notched-fatigued specimens. Both models have positive CSA coefficients, indicating that an increase in cross-sectional area leads to an increase in elongation at break. Even so, the coefficient in the ε_n_6061-T4_ model has a slightly greater value, therefore an increased impact of the cross-sectional area on elongation for notched specimens. The sign of the interaction term varies between the two models. The ε_n_6061-T4_ model’s outcome is negative, indicating that the combined effect of NDir and CSA has a lesser influence on elongation. The ε_nf_6061-T4_ model results in a favourable result, indicating a minor but perceptible increase in impact. It exhibits quadratic components for NDir and CSA, as well, indicating a non-linear relationship between these variables and elongation at break. This implies that the behaviour of notched-fatigued specimens is complex.

For aluminium alloy 6061-T4, the variations in the models suggest that fatigue significantly impacts the relationship between notch direction, cross-sectional area, and elongation at break.
ε_n_6061-T4_ = 6.1297 − 0.083523 × NDir + 0.215888 × CSA − 0.00015 × NDir × CSA,(20)
ε_nf_6061-T4_ = 5.48139 − 0.083667 × NDir + 0.183291 × CSA + 0.000227 × NDir × CSA + 0.00012 × NDir^2^ − 0.000647 × CSA^2^.(21)

Figure 16a’s contour map of notched specimens highlights the relationship between elongation at break and the direction of the notch. As the notch angle increases, there is a consistent reduction in elongation. In this case, the cross-sectional area has a less significant influence on elongation.

Figure 16b is the contour map of the notched-fatigued specimens, indicating the complexity and correlation between the variables. Despite the noticeable effect of notch direction on elongation, its influence appears to be less significant compared to the notched specimens. A stronger tendency is observed for the cross-sectional area, with a noticeable decrease in elongation as the cross-sectional area increases, as indicated by the downward slope. The results indicate that fatigue significantly reduces the material’s capacity to elongate, and the impact of the cross-sectional area becomes even more important when considering the extent to which the material may stretch before breaking in these circumstances.

Upon comparing the two graphs, it can be noted that, in the case of aluminium alloy 6061-T4, the fatigued specimen’s mechanical characteristics decrease significantly. The plot in Figure 16a has significantly lower overall elongation values, as evidenced by the colour scale shifting toward cooler hues. In addition, the impact of notch direction and cross-sectional area varies between the two conditions, with the cross-sectional area having a more significant effect in the case of notched-fatigued specimens.

On average, the elongation at break of the fatigued specimens of 6061-T4 uncoated aluminium alloy decreases by up to 56.057% compared to the notched specimens. The results indicate a decrease of 237.406% (notched specimens) and 344.524% (notched-fatigued) compared to the initial values. It can be noted that for this alloy, low values for elongation at break were measured at 0.532% to 90°, with a cross-sectional area of 12.5 mm^2^, notched-fatigued.

In the case of 6061-T4 uncoated aluminium alloy, the ε_nf_6061-T4_uncoated_ model from Equation (23) has an increased intercept value (5.79566) in contrast to ε_n_6061-T4_uncoated_ (5.34183), indicating a generally higher baseline elongation for notched-fatigued specimens.

Both models’ (Equations (22) and (23)) NDir coefficients are negative, indicating that increasing the notch direction angle correlates to a decrease in elongation at break. However, the effect is much stronger in the ε_nf_6061-T4_uncoated_ model, showing a stronger link between the direction of the notch and the lengthening of specimens that have been notched-fatigued.

In both models, the CSA coefficient is positive, suggesting that an increase in cross-sectional area typically leads to an increase in elongation at break. Even so, the coefficient in ε_n_6061-T4_uncoated_ is higher, which means that the cross-sectional area has an increased impact on elongation for notched specimens.

The interaction term’s sign varies between the two models. The results of the ε_n_6061-T4_uncoated_ model led to a negative relationship, indicating that the combined action of NDir and CSA results in decreasing the elongation at break. ε_nf_6061-T4_uncoated_ leads to a favourable result, indicating a minor but noticeable increased impact.

The ε_nf_6061-T4_uncoated_ model has quadratic factors for NDir and CSA, suggesting a non-linear correlation between these parameters and elongation at break.
ε_n_6061-T4_uncoated_ = 5.34183 − 0.077772 × NDir + 0.19612 × CSA − 0.00015 × NDir × CSA,(22)
ε_nf_6061-T4_uncoated_ = 5.79566 − 0.088381 × NDir + 0.119279 × CSA + 0.000227 × NDir × CSA + 0.00012 × NDir^2^ − 0.000647 × CSA^2^.(23)

The contour map of notched aluminium alloy 6061-T4 uncoated (Figure 17a) indicates that as the notch angle increases, there is a gradual decrease in elongation, as seen by the transition from warmer to cooler hues. This indicates that the direction of the notch has a major effect on the material’s ability to withstand plastic deformation without breaking. The influence of cross-sectional area on elongation tends to be less significant, however, for larger cross-sectional areas. The notched-fatigued aluminium alloy 6061-T4 uncoated contour map (Figure 17b) reveals a more complex relationship among the variables, which aligns with the mathematical model from Equation (23). While the impact of the notch direction on elongation remains noticeable, it seems to be less significant compared to the notched specimens. A distinct pattern can be noted, indicated by the downward slope of the colour bands. Under these circumstances, fatigue significantly reduces the material’s ability to stretch, and the impact of the cross-sectional area becomes more important in determining the extent of elongation at the point of fracture.

When comparing the two plots, fatigue has a significant negative impact on the mechanical characteristics of the uncoated aluminium alloy 6061-T4. The elongation values in Figure 17b are much lower. In addition, the direction of the notch and the cross-sectional area have different effects on the two factors, with the cross-sectional area having a bigger effect on notched-fatigued specimens.

The statistics given emphasise the significance of considering both the direction of scratches and the cross-sectional area when assessing the mechanical properties of uncoated aluminium alloy 6061-T4, particularly in the context of fatigue.

Notched-fatigued specimens, in the case of aluminium alloy 6061-T6 uncoated, are 57.664% likely to fail compared to notched specimens and 251.917% when compared to the initial properties for the elongation. A 16.65% average percentage difference was measured between the notched and notched-fatigue specimens.

The intercept of Equation (25) for ε_nf_6061-T6_uncoated_ is 4.55938, which is less than the intercept for the ε_n_6061-T6_uncoated_ model, which is 6.84834, from Equation (24). This implies that notched-fatigued specimens typically exhibit a lower elongation at break.

The NDir coefficient for both models is negative, showing that a decrease in notch direction can result in an increase in elongation at break. Still, the magnitude is significantly greater in the ε_n_6061-T6_uncoated_ model, indicating a more significant effect of notch direction on elongation for the notched specimens.

In both models, the CSA coefficient is positive, indicating that an increase in cross-sectional area usually leads to an increase in elongation at break. However, the coefficient in the ε_nf_6061-T6_uncoated_ model is higher, meaning that the cross-sectional area impact on elongation for notched-fatigued specimens is higher.

The interaction term’s sign varies between the two models. In the ε_n_6061-T6_uncoated_ model, the coefficient is negative, indicating the decreasing impact of the combined NDir and CSA on elongation. The ε_nf_6061-T6_uncoated_ model provides an improved outcome, indicating a minor upward impact. The ε_nf_6061-T6_uncoated_ model has quadratic factors for NDir and CSA, which means that these parameters do not have a straight-line relationship with elongation at break. This implies complex behaviour in notched-fatigued specimens. The differences between the models for uncoated aluminium alloy 6061-T6 show that fatigue has a big impact on the relationship between notch direction, cross-sectional area, and elongation at break.
ε_n_6061-T6_uncoated_ = 6.84834 − 0.074378 × SDir + 0.046334 × CSA − 0.00015 × SDir × CSA,(24)
ε_nf_6061-T6_uncoated_ = 4.55938 − 0.067235 × SDir + 0.106653 × CSA + 0.000227 × SDir × CSA + 0.00012 × SDir^2^ − 0.000647 × CSA^2^.(25)

The angle of the notch affects the elongation at break for notched specimens of the uncoated aluminium alloy 6061-T6 (Figure 18a), as shown for all alloys, as the angle of the notch increases, the elongation decreases. The effect of cross-sectional area on elongation tends to be less significant; however, there is a minor increase in elongation for larger cross-sectional areas, as expected.

In Figure 18b, the notched-fatigued specimens highlight a complex correlation between the variables. Despite the noticeable effect of notch direction on elongation, the results show less significance compared to the notched specimens.

The comparison of both plots leads to the conclusion that fatigue significantly impairs the mechanical characteristics of the uncoated aluminium alloy 6061-T6.

The mathematical models from Equations (24) and (25) depict the correlation between elongation at break and notch direction (NDir), as well as the cross-sectional area (CSA). In all models, the key predictors of elongation at break are a linear combination of NDir, CSA, and their interactions. Each alloy’s sf mode exhibits quadratic factors for both NDir and CSA, suggesting a more complex relationship between these variables and elongation at break in both notched-fatigued specimens. The intercept values show substantial variation, suggesting that there are intrinsic variances in ductility across the alloys. In addition, the NDir, CSA, and interaction terms show different values, showing complicated responses to notch direction and cross-sectional area for each alloy.

Among the many types of alloys, the models for notched-fatigued specimens exhibit significant differences. Notched-fatigued specimens have an overall tendency to decrease elongation, given a lower intercept. The coefficients for NDir and CSA show variation, indicating that fatigue has modified sensitivity to these parameters. Using quadratic factors in the models for notched-fatigued specimens also shows that the direction of the notch and the area of the notch do not have a straight-line effect on the lengthening when the material is fatigued.

#### 3.2.3. Aluminium Alloys 7075-T0 and 7075-T6

The result of the aluminium alloy 7075-T0 indicates that on average, specimens notched at 0° fail at 9.586% elongation, compared to 7.861% when subjected to fatigue, meaning a decrease of 21.952%. This tendency is also noted when analysing the specimens notched at 45° (from 5.869% to 4.945%, resulting in a decrease of 18.7%) and 90° (from 1.81% to 1.545%, resulting in a decrease of 17.137%).

When looking at the aluminium alloy 7075-T0 ε_nf_7075-T0_ model from Equation (27), it has a lower intercept value (5.74385) than the ε_n_7075-T0_ model (6.06584) from Equation (26). This means that specimens that are notched and worn out have a lower elongation at break. The NDir coefficient for both models is negative, suggesting that an increase in the notch direction results in reduced values for the elongation at break. The value is somewhat greater in the ε_nf_7075-T0_ model, indicating a more significant effect of notch direction on elongation for notched-fatigued specimens.

Both models have positive CSA coefficients, indicating that an increase in the cross-sectional area leads to an increase in elongation at break. Nevertheless, the coefficient in the ε_n_7075-T0_ model is higher, showing a greater influence of cross-sectional area on elongation for notched specimens.

The interaction term’s sign varies between the two models. In the ε_n_7075-T0_ model, the coefficient is negative, indicating the decreasing impact of the combined NDir and CSA on elongation. The ε_nf_7075-T0_ model provides a favourable result, indicating a minor but noticeable increase. The ε_nf_7075-T0_ model has quadratic factors for NDir and CSA, suggesting a non-linear correlation between these parameters and elongation at break.
ε_n_7075-T0_ = 6.06584 − 0.070452 × NDir + 0.17151 × CSA − 0.00015 × NDir × CSA,(26)
ε_nf_7075-T0_ = 5.74385 − 0.073863 × NDir + 0.1238097 × CSA + 0.000227 × NDir × CSA + 0.00012 × NDir^2^ − 0.000647 × CSA^2^.(27)

The same tendency can be noted, from Figure 19a, in the case of notched aluminium alloy 7075-T0 specimens: As the angle of the notch increases and the cross-section area decreases, there is a consistent pattern of decreasing elongation, affecting the alloy’s ductility. 

For notched-fatigued specimens, Figure 19b indicates the correlation between the variables, as seen in the previous cases. Although the effect of notch direction on elongation remains noticeable, it seems to be less significant compared to the notched specimens. The same correlation can be observed between the cross-sectional area and elongation, with a decrease in elongation as the cross-sectional area decreases. Comparing the two plots reveals a substantial negative impact of fatigue on the mechanical properties of the aluminium alloy 7075-T0, with the cross-sectional area having a more significant effect in the case of notched-fatigued specimens.

In the case of aluminium alloy 7075-T6, it can be noted that, on average, notched specimens fail at 4.422%, while notched-fatigued ones at 3.551%; this leads to a decrease of 24.54% in elongation at break. Notched specimens show an average decrease of 216.574% compared to the initial elongation value; applying fatigue bending reduces these values to 294.261%.

For the aluminium alloy 7075-T6, the ε_nf_7075-T6_ model has a lower intercept value of 3.29608, indicated in Equation (29), in contrast to ε_n_7075-T6_’s 4.34604, from Equation (28), indicating a reduced elongation at break for notch and fatigued specimens.

Both models resulted in negative NDir coefficients, suggesting that an increase in notch direction results in a decrease in elongation at break. The effect is somewhat greater in the ε_nf_7075-T6_ model, indicating a more significant impact of notch direction on elongation for notched-fatigued specimens. Positive CSA coefficients indicate that an increase in cross-sectional area leads to an increase in elongation at break. In the ε_nf_7075-T6_ model, the coefficient has a slightly larger scale, indicating an increased effect of cross-sectional area on elongation for notched-fatigued specimens. The sign of the interaction term differs between the two models. The ε_n_7075-T6_ model has a negative relationship, indicating that the combined NDir and CSA have a decreasing impact on elongation. The ε_nf_7075-T6_ model provides a favourable result, indicating a minor but notable improvement.

The ε_nf_7075-T6_ model also has quadratic factors for NDir and CSA, indicating a non-linear correlation between these variables and elongation at break. As anticipated, this implies a more complex behaviour in notched-fatigued specimens.
ε_n_7075-T6_ = 4.34604 − 0.046605 × NDir + 0.112898 × CSA − 0.00015 × NDir × CSA,(28)
ε_nf_7075-T6_ = 3.29608 − 0.049071 × NDir + 0.116246 × CSA + 0.000227× NDir × CSA + 0.00012 × NDir^2^ − 0.000647 × CSA^2^.(29)

The contour map from Figure 20a of notched 7071-T6 specimens highlights the relationship between elongation at break and the direction of the notch. As the notch angle increases, there is a consistent pattern of decreasing elongation. Lower values of the cross-section area led to low elongation at break values. In Figure 20b, the data for notched-fatigued 7075-T6 specimens shows a more complex correlation between the variables. Despite being noticeable, the influence of the notch direction on elongation appears to be less significant compared to the notched specimens. A noticeable pattern is seen in the cross-sectional area, where elongation reduces noticeably as the cross-sectional area decreases. As previously concluded, fatigue significantly reduces the material’s ductility, and the impact of the cross-sectional area becomes more important in determining the extent of elongation at the fracture point under these circumstances.

Fatigue has a substantial negative impact on the mechanical properties of the aluminium alloy 7075-T6. Figure 20b’s plot shows significantly reduced overall elongation values. In addition, the cross-sectional area results have a relevant effect on notched-fatigued specimens. The results indicate that these parameters are important in reducing the material’s performance in applications that experience repeated stress and surface deterioration.

## 4. Discussion

The analysis of variance (ANOVA) establishes mathematical models that characterize the relationship between UTS, elongation at break, and other characteristics such as notch direction and cross-sectional area. Quadratic or 2FI interference models examined the data with each model yielding a *p*-value below 0.001, indicating results that are statistically significant. A non-linear relationship between the direction of the notch and the cross-sectional area was identified; this correlation was shown to affect the strength and elongation of the alloys. Fatigue has a significant impact on the mechanical characteristics of the aluminium alloy, resulting in a decrease in the UTS and elongation at break. The values continue to drop as the notch direction and cross-sectional area decrease.

The interpretation of the results can be summarized in the data presented in Table 7, where the initial values for UTS and elongation at break are presented, together with the average values obtained for the notched (UTS_n_avg_, ε_n_avg_) and notched-fatigued (UTS_nf_avg_, ε_nf_avg_) specimens; the percentage difference (% diff.) between these and the initial values is also presented.

When analysing the UTS_n_avg_, aluminium alloys 2024-T3 and 7075-T6 withstand scratches much better, with a decrease of 5.17% and 4.151%, respectively. On the other side are the 7075-T0 and 6061-T4 alloys, which decreased by 10.8% and 11.92%, respectively. Inducing fatigue in the process alters this order, with the 6061-T6 uncoated alloy achieving superior results (28.05%), while the 6061-T4 uncoated alloy falls at the opposite end (41.6%). Thus, the data indicate a decrease between 20.78% and 30.26% in the UTS when fatigue is applied to a notched specimen.

The data indicate that 7075-T6 exhibits the highest initial UTS, followed by 2024-T3, 6061-T4, and the uncoated alloys. This suggests that alloy composition and heat treatment significantly influence the material’s strength. Both notch and fatigue lead to a substantial reduction in UTS for all alloys. The percentage decrease in UTS varies among alloys, with 6061-T4 uncoated experiencing the most significant drop.

Table 7 also notes significant variations in elongation behaviour among the investigated aluminium alloys, with 6061-T4 exhibiting the highest initial elongation (25%), followed by 2024-T3 (16%), 7075-T0 (15%), and 6061-T6 and 7075-T6 uncoated at 12.8% and 14%, respectively. The percentage difference provides quantitative insights into the percentage reduction in elongation due to the notch and fatigue. The 6061-T4 alloy experiences a 310.94% reduction in elongation due to the notch, while the reduction due to fatigue is 385.64%. With notches, the elongation decrease of the 2024-T3 alloy is 154.15% in this case, and after fatigue, it is 210.63%, indicating that it possesses a highly desirable behaviour.

It can be noted that the effect of notches and subsequent fatigue have a negative impact on the ultimate tensile strength. The UTS values for both notched and notched-fatigued conditions consistently show considerable decreases relative to the initial UTS values for all the alloys examined.

Uncoated alloy 6061-T4 exhibits a major decrease in UTS when subjected to both notch and fatigue, suggesting an increased susceptibility to strength loss in comparison to other alloys.

A potential correlation between the initial UTS and the extent of UTS decrease due to notch and fatigue. Alloys that have greater initial UTS values, such as 7075-T6, show less significant decreases compared to alloys with lower initial UTS values, such as 6061-T4 uncoated. This suggests that alloys with higher initial strength may exhibit greater resistance to deterioration caused by notch and fatigue.

Notching and fatigue cause a substantial reduction in elongation compared to the initial values for all aluminium alloys. This indicates that these factors have a negative impact on the material’s ductility. The percentage difference in elongation between the initial state and the notched or fatigued conditions varies across different alloys. While all the alloys experience a decrease in elongation, the magnitude of this reduction is not uniform.

According to the provided data, alloy 7075-T6 exhibits superior results in UTS and elongation. The 7075-T6 alloy has the highest initial UTS (530 MPa), followed by 6061-T6 (310 MPa), 2024-T3 (450 MPa), and 6061-T4 (255 MPa and 247 MPa). While all the alloys experience a reduction in UTS due to notch and fatigue, 7075-T6 maintains the highest UTS even under these adverse conditions. The 6061-T4 alloy shows the highest initial elongation (25% and 22%), indicating superior ductility compared to other alloys. However, for 6061-T4, the percentage reduction in elongation due to notch and fatigue is also significant. 7075-T6, despite having a lower initial elongation, shows superior resistance to elongation loss under notch and fatigue conditions. While 6061-T4 exhibits higher initial ductility, 7075-T6 offers a superior balance of strength and ductility, especially when considering the effects of notch and fatigue. Its ability to maintain relatively high UTS and elongation under adverse conditions makes it a more suitable choice for applications demanding both strength and toughness. 

Although it outperforms other alloys in terms of both UTS and elongation, the aluminium alloy 2024-T3 shows a balanced combination of strength and ductility. While it exhibits a reasonable initial UTS, its susceptibility to strength degradation under notch and fatigue conditions is notable. The UTS of 2024-T3 situates itself between the higher strength of 7075-T6 and the lower strength of 6061-T4 alloys, while the elongation values fall between the higher ductility of 6061-T4 and the lower ductility of 7075-T6.

While 2024-T3 offers a reasonable balance of strength and ductility, its susceptibility to degradation under adverse conditions limits its suitability for applications demanding high levels of resistance to notches and fatigue. For applications requiring optimal strength, 7075-T6 might be a more suitable choice, whereas for applications prioritising ductility, 6061-T4 could be considered.

## 5. Conclusions

This research study aimed to investigate the effects of induced scratches on the UTS and elongation at break for aircraft-grade aluminium alloys subjected to low-cycle fatigue bending, as well as to establish an overview of the factors that contribute to material failure and consequent loss of structural integrity.

To choose the appropriate aluminium alloy for a particular application, it is essential to consider the predicted loading conditions, as well as the possibility of surface damage.

Of course, to fully understand the phenomenon, one must take into account the following aspects and explore potential future research directions: analysis of the microstructure (changes in microstructure that occur during notch and fatigue), initiation and propagation of cracks (examine the impact of notch shape and position on the way fatigue cracks develop or the correlation between multiple scratches of different orientations and sizes), application of finite element analysis (FEA) to model the distribution of stress and strain around scratches and forecast the beginning and propagation of cracks or examine the effects of multi-axial loading, such as tension–torsion and bending–torsion, on the mechanical properties of aluminium alloys that have been notched-fatigued.

## Figures and Tables

**Figure 1 materials-17-04639-f001:**
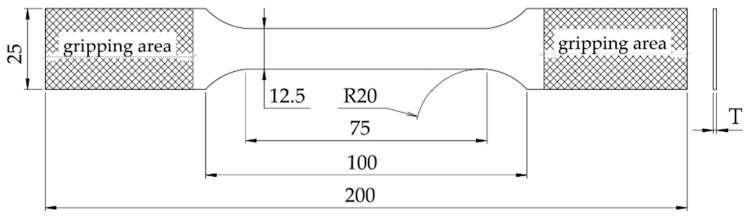
Aluminium alloy specimen dimensions in millimetres, indicating with T the specimen thickness (1.0, 1.2, 1.27, 1.6, 1.8, 2) and gripping areas.

**Figure 2 materials-17-04639-f002:**
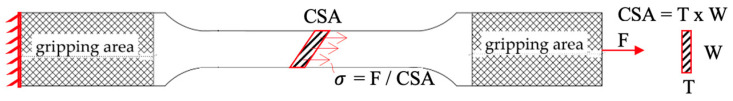
Schematic representation of the tensile test, indicating the gripping area, fixture, applied force direction (F), cross-section area (CSA), and internal stress (σ).

**Figure 3 materials-17-04639-f003:**
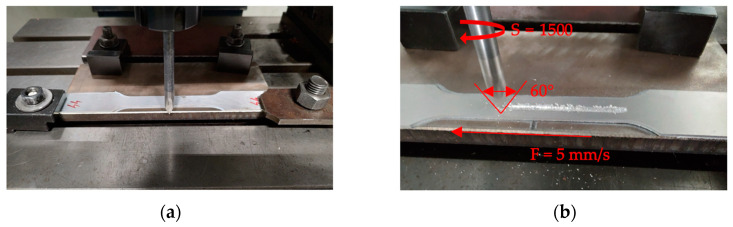
Notch execution process of the tensile test specimens: (**a**) specimen mounted on the milling machine for scratching; (**b**) close-up view of the scratching tool with the corresponding parameters.

**Figure 4 materials-17-04639-f004:**
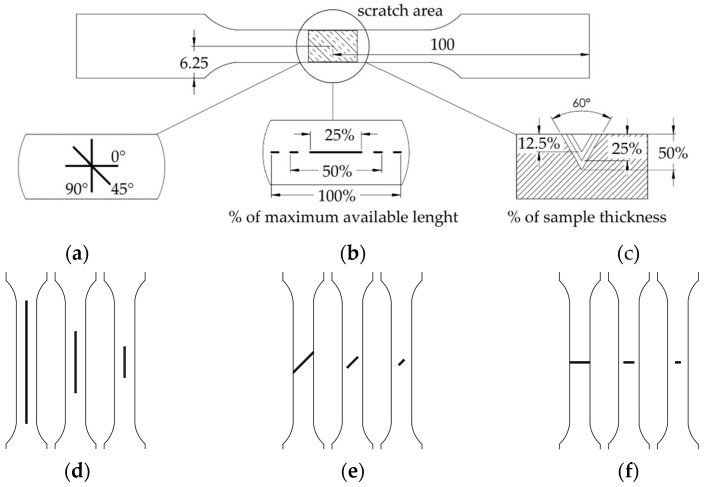
Schematic representation of specimen geometry, notch area and dimensions: (**a**) notch orientation (0°, 45°, 90°); (**b**) notch length, as % of maximum available length, depending on the notch angle; (**c**) notch depth, as % of specimen thickness; (**d**) 0° notches, with lengths of 18.75, 35, and 75 mm; (**e**) 45° notch, with lengths of 4.419, 8.839, and 17.678 mm; (**f**) 90° notch, with lengths of 3.125, 6.25, and 12.5 mm.

**Figure 5 materials-17-04639-f005:**
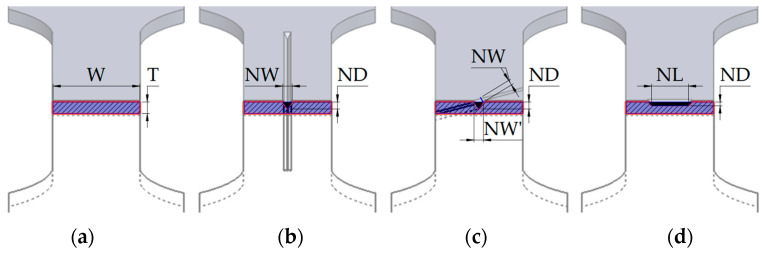
Transversal section view illustrating the cross-sectional dimensions of the aluminium specimens. (**a**) Standard specimen (W—specimen width and T—specimen thickness); (**b**) specimen with 0° notch (NW—notch width and ND—notch depth); (**c**) specimen with 45° notch (NW—notch width, ND—notch depth, and NW’—transversal projection of NW); (**d**) specimen with 90° notch (NL—notch length and ND—notch depth).

**Figure 6 materials-17-04639-f006:**
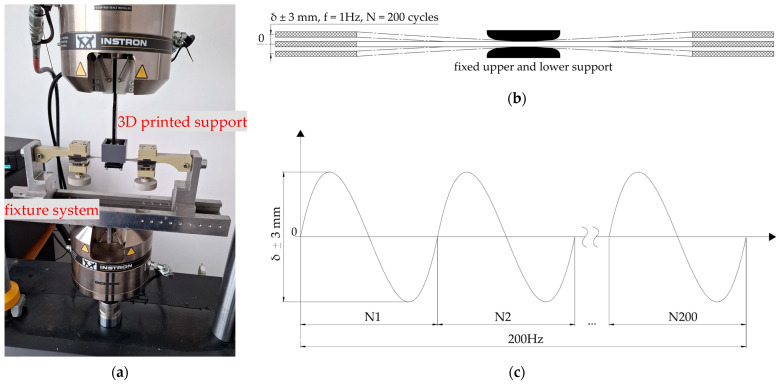
Fatigue bending experimental setup and process parameters: (**a**) fatigue equipment with fixture system and 3D printed support; (**b**) schematic representation of the fatigue cycle process; (**c**) fatigue cycle parameters plot.

**Figure 7 materials-17-04639-f007:**
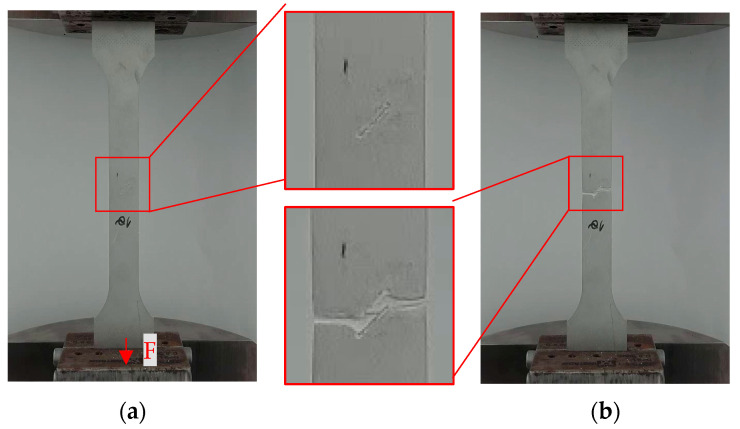
Tensile test setup indicating: (**a**) start of the process and load direction with close-up on the notch and (**b**) end of the process with close-up on the crack propagation resulting in specimen breaking.

**Figure 8 materials-17-04639-f008:**
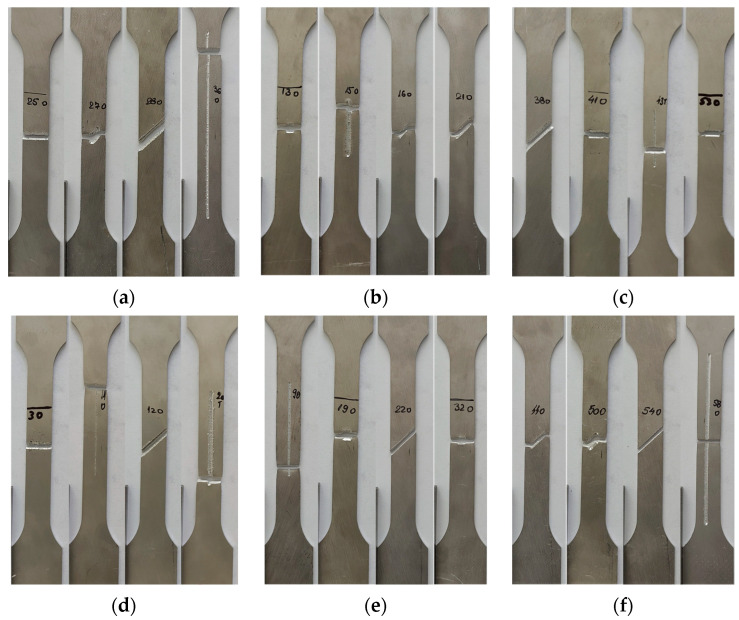
Examples of crack propagation resulting in specimen breaking: (**a**) 2024-T3; (**b**) 6061-T4; (**c**) 6061-T4 uncoated; (**d**) 6061-T6 uncoated; (**e**) 7071-T0; (**f**) 7071-T6.

**Figure 9 materials-17-04639-f009:**
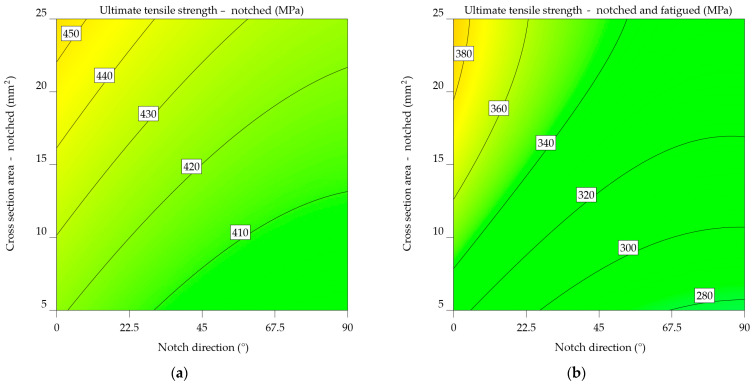
Contour plots indicating the effect of NDir and CSA on the UTS for aluminium alloy 2024-T3, with warmer colours representing higher UTS values: (**a**) data visualization for notched specimens; (**b**) data visualization for notched-fatigued specimens.

**Figure 10 materials-17-04639-f010:**
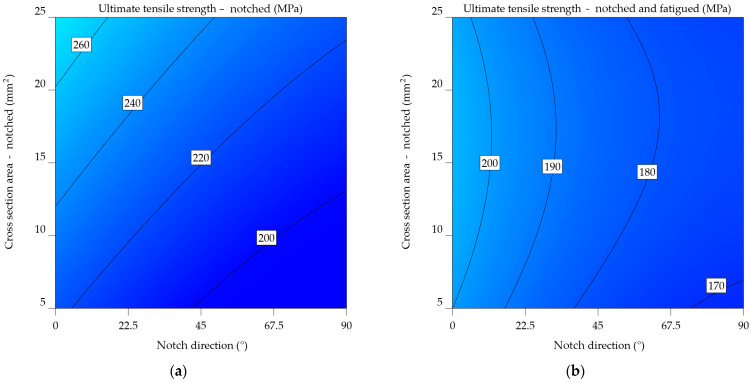
Contour plots indicating the effect of NDir and CSA on the UTS for aluminium alloy 6061-T4, with warmer colours representing higher UTS values: (**a**) data visualization for notched specimens; (**b**) data visualization for notched-fatigued specimens.

**Figure 11 materials-17-04639-f011:**
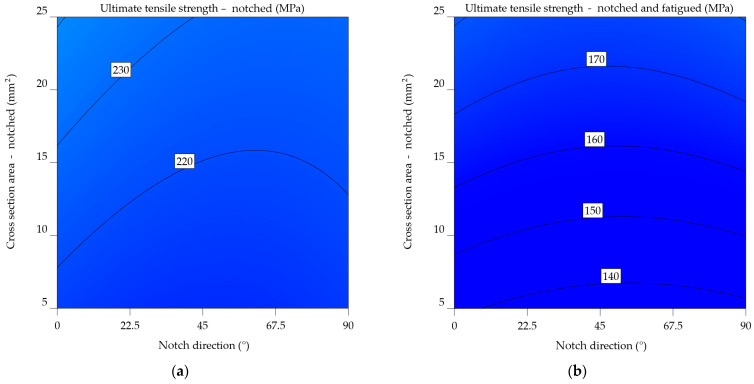
Contour plots indicating the effect of NDir and CSA on the UTS for aluminium alloy 6061-T4 uncoated, with warmer colours representing higher UTS values: (**a**) data visualization for notched specimens; (**b**) data visualization for notched -fatigued specimens.

**Figure 12 materials-17-04639-f012:**
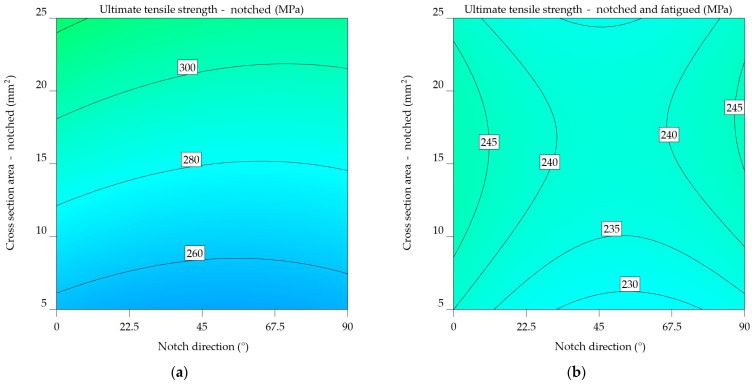
Contour plots indicating the effect of NDir and CSA on the UTS for aluminium alloy 6061-T6 uncoated, with warmer colours representing higher UTS values: (**a**) data visualization for notched specimens; (**b**) data visualization for notched-fatigued specimens.

**Figure 13 materials-17-04639-f013:**
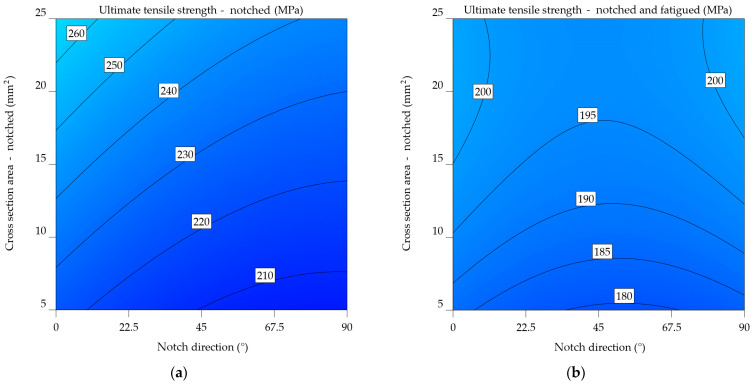
Contour plots indicating the effect of NDir and CSA on the UTS for aluminium alloy 7075-T0, with warmer colours representing higher UTS values: (**a**) data visualization for notched specimens; (**b**) data visualization for notched-fatigued specimens.

**Figure 14 materials-17-04639-f014:**
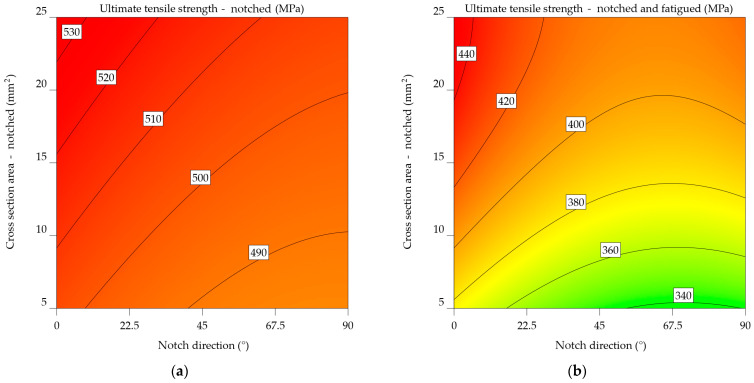
Contour plots indicating the effect of NDir and CSA on the UTS for aluminium alloy 7075-T6, with warmer colours representing higher UTS values: (**a**) data visualization for notched specimens; (**b**) data visualization for notched-fatigued specimens.

**Figure 15 materials-17-04639-f015:**
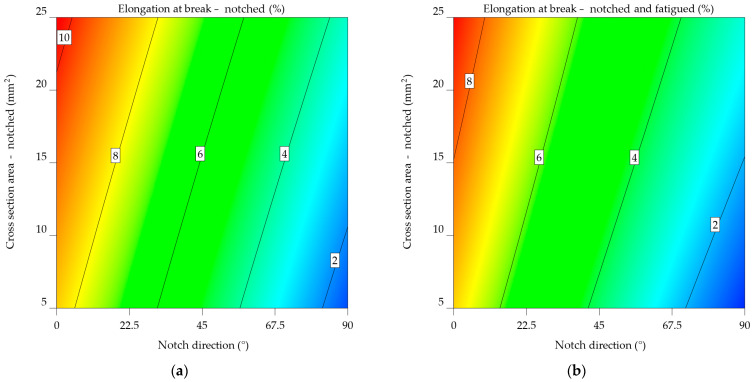
Contour plots indicating the effect of NDir and CSA on the elongation at break for aluminium alloy 2024-T3, with warmer colours representing higher elongation values: (**a**) data visualization for notched specimens; (**b**) data visualization for notched-fatigued specimens.

**Figure 16 materials-17-04639-f016:**
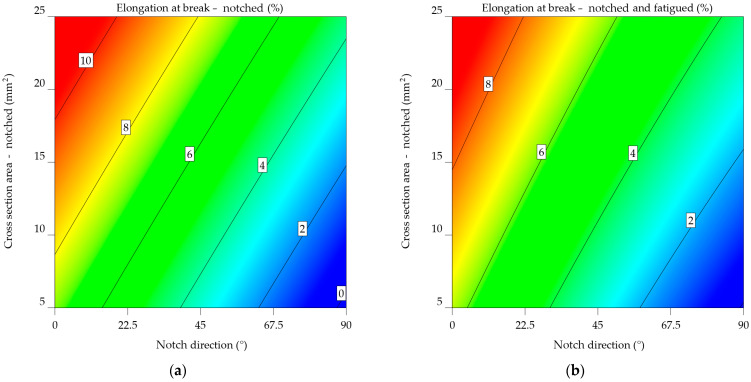
Contour plots indicating the effect of NDir and CSA on the elongation at break for aluminium alloy 6061-T4, with warmer colours representing higher elongation values: (**a**) data visualization for notched specimens; (**b**) data visualization for notched-fatigued specimens.

**Figure 17 materials-17-04639-f017:**
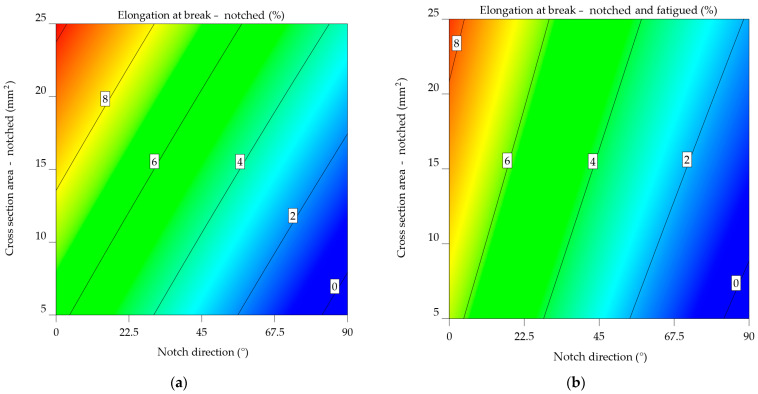
Contour plots indicating the effect of NDir and CSA on the elongation at break for aluminium alloy 6061-T4 uncoated, with warmer colours representing higher elongation values: (**a**) data visualization for notched specimens; (**b**) data visualization for notched-fatigued specimens.

**Figure 18 materials-17-04639-f018:**
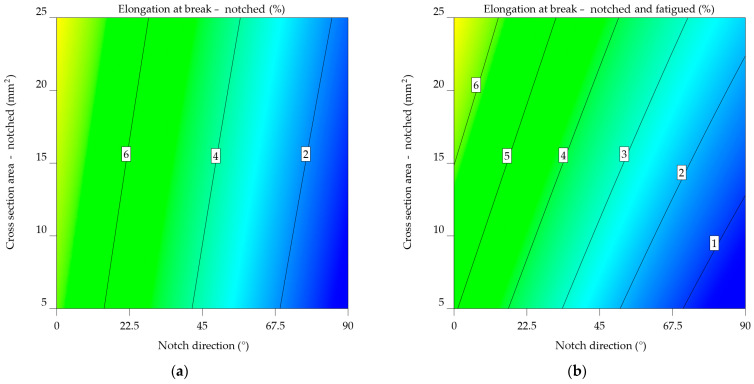
Contour plots indicating the effect of NDir and CSA on the elongation at break for aluminium alloy 6061-T6 uncoated, with warmer colours representing higher elongation values: (**a**) data visualization for notched specimens; (**b**) data visualization for notched-fatigued specimens.

**Figure 19 materials-17-04639-f019:**
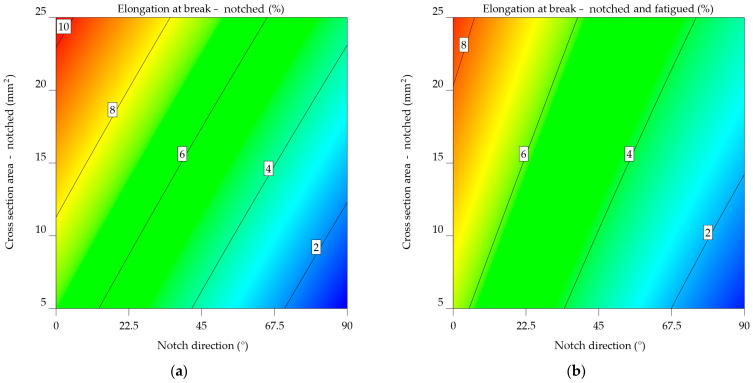
Contour plots indicating the effect of NDir and CSA on the elongation at break for aluminium alloy 7075-T0, with warmer colours representing higher elongation values: (**a**) data visualization for notched specimens; (**b**) data visualization for notched-fatigued specimens.

**Figure 20 materials-17-04639-f020:**
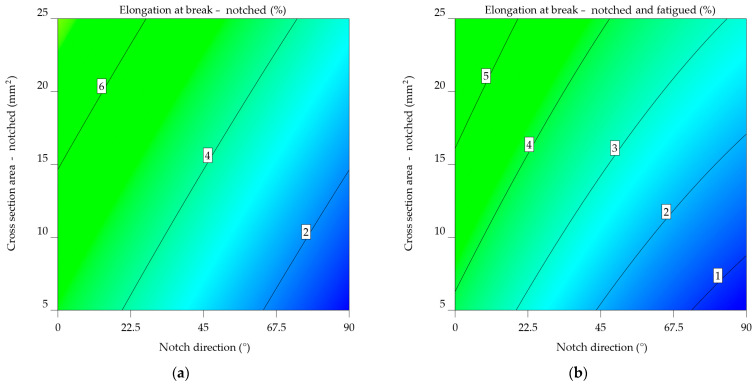
Contour plots indicating the effect of NDir and CSA on the elongation at break for aluminium alloy 7075-T6, with warmer colours representing higher elongation values: (**a**) data visualization for notched specimens; (**b**) data visualization for notched-fatigued specimens.

**Table 1 materials-17-04639-t001:** Aluminium alloys tensile test mechanical properties.

Aluminium Alloy	Youngs Modulus (GPa)	Yield 0.2% (MPa)	Ultimate Tensile Strength (MPa)	Elongation at Break (%)
2024-T3	73.1	291	450	16
6061-T4	68.9	165	255	25
6061-T4 uncoated	68.9	138	247	22
6061-T6 uncoated	68.9	281	310	12.8
7075-T0	71.7	131	260	15
7075-T6	70	487	530	14

**Table 2 materials-17-04639-t002:** Experimental plan data, factor type, factor description, and levels used in this study.

Data Type	Factor Type	Factor Description	Levels
Input	Numeric, discrete factor	Specimen thickness T (mm)	1, 1.2, 1.27, 1.6, 1.8, 2
		Notch direction (°)	0, 45, 90
		Cross-section area CSA (mm^2^)	Complete description in Table 3
	Categoric, nominal factor	Specimen material	2024-T3, 6061-T4, 6061-T4 uncoated, 6061-T6 uncoated, 7075-T0, 7075-T6
Output	Response, analysed as polynomial	UTS_n_ (MPa), UTS_nf_ (MPa)	-
		ε_n_ (%), ε_nf_ (%)	-

**Table 4 materials-17-04639-t004:** Ultimate tensile strength (UTS_n_—notched and UTS_nf_—notched-fatigued) and elongation at break (ε_n_—notched and **ε**_nf_—notched-fatigued), along with the corresponding CSA and notch direction for each aluminium alloy.

Aluminium Alloy	Notch Direction (°)	Cross-Section Area (mm^2^)	UTS_n_ (MPa)	ε_n_ (%)	UTS_nf_ (MPa)	ε_nf_ (%)
2024-T3	0	12.49	435.50	9.14	358.89	7.81
		12.36	426.16	8.92	355.02	7.74
		24.96	459.73	10.31	400.99	8.84
		19.98	443.72	9.55	374.98	8.13
		19.98	448.03	9.68	379.46	8.14
		22.03	454.95	9.95	392.41	8.56
	45	12.49	418.66	6.21	318.42	4.65
		14.93	424.71	6.79	316.76	4.98
		24.18	426.52	7.01	325.62	5.40
		24.18	427.52	7.26	346.36	5.74
	90	14.01	411.95	2.14	322.52	1.88
		13.13	406.89	1.81	299.93	1.49
		21.77	422.48	3.11	321.08	2.45
		16.88	412.13	2.51	323.83	2.13
		17.41	419.45	2.82	332.34	2.34
6061-T4	0	19.63	269.12	10.38	212.61	8.62
		15.64	244.44	9.94	202.88	8.65
		15.86	248.72	9.95	206.44	8.65
		15.86	256.16	9.93	217.73	8.44
	45	19.97	232.87	6.98	188.63	5.79
		19.87	227.83	6.60	184.54	5.48
		19.48	224.98	6.34	184.48	5.14
		15.55	203.26	3.80	166.68	3.23
	90	18.66	217.63	3.87	178.45	3.25
		7.94	195.61	0.67	176.05	0.64
6061-T4 uncoated	0	24.96	239.18	9.93	181.77	7.75
	45	24.18	232.13	7.36	171.78	6.07
	90	23.40	228.43	3.16	181.32	1.50
		21.73	223.22	2.64	171.88	1.08
		12.50	220.27	1.00	156.04	0.53
6061-T6 uncoated	0	19.98	305.88	7.85	250.82	6.43
		19.63	298.12	7.49	244.46	6.22
		19.98	314.22	7.98	248.23	6.62
	45	19.48	293.67	4.52	237.87	3.79
	90	10.00	267.44	0.76	240.70	0.69
7075-T0	0	19.98	257.11	9.81	210.83	8.04
		19.91	253.07	9.37	202.46	7.68
	45	19.87	234.46	6.19	192.88	5.14
		19.97	244.87	6.66	193.45	5.46
		12.45	221.27	4.76	180.29	4.24
	90	17.13	221.07	2.79	199.91	2.00
		10.79	215.79	1.94	196.37	1.77
		11.71	219.07	1.87	194.97	1.66
		6.25	208.57	0.89	186.14	0.86
		9.38	212.00	1.56	192.92	1.44
7075-T6	0	22.38	526.02	7.53	401.78	5.70
		24.96	528.12	7.06	399.37	5.89
		15.64	519.02	5.83	446.36	4.96
	45	15.85	508.77	4.21	422.28	3.58
		19.97	510.76	4.38	422.39	3.59
		22.34	513.81	4.65	434.50	3.68
		22.34	514.91	4.70	435.78	3.76
		12.49	502.16	3.79	381.85	2.45
		24.18	516.23	5.13	445.34	3.95
		12.49	503.66	3.85	378.26	2.62
	90	10.00	481.88	1.11	362.87	1.01
		17.13	478.49	3.00	387.58	2.49
		10.94	485.88	1.31	350.57	1.18
		21.88	508.06	2.65	386.12	2.10
		13.89	506.74	2.29	381.93	2.04

**Table 5 materials-17-04639-t005:** Model fit statistics and transformation parameters for each studied parameter.

Fit Statistics	UTS_n_	ε_n_	UTS_nf_	ε_nf_
Model	Quadratic	2FI	Quadratic	Quadratic
Model *p*-value	<0.001	<0.001	<0.001	<0.001
Box–Cox transformation	λ = 1, none	λ = 1, none	λ = −0.5, inverse sqrt	λ = 1, none

**Table 6 materials-17-04639-t006:** Goodness-of-fit metrics; model fit statistics for predicting UTS_n_, UTS_nf_, ε_n,_ and ε_nf_.

Fit Statistics	UTS_n_	ε_n_	UTS_nf_	ε_nf_
R^2^	0.9277	0.9115	0.8990	0.8804
Adjusted R^2^	0.9266	0.8934	0.8734	0.8703
Predicted R^2^	0.8743	0.8108	0.8966	0.8904
Adequate Precision	82.8293	85.3889	142.2185	91.4676

**Table 7 materials-17-04639-t007:** Outline the experimental data, including the average values compared to the initial values and the percentage decrease for each alloy.

Aluminium Alloy	UTS_initial_ (MPa)	UTS_n_avg_		UTS_nf_avg_		ε_initial_ (%)	ε_n_avg_		ε_nf_avg_	
		MPa	% diff.	MPa	% diff.		%	% diff.	%	% diff.
2024-T3	450	427.87	5.17	341.23	31.87	16.00	6.30	154.15	5.15	210.63
6061-T4	255	227.82	11.92	189.41	34.62	25.00	6.08	310.94	5.14	385.64
6061-T4 uncoated	247	231.76	6.57	174.43	41.60	22.00	6.52	237.40	4.95	344.52
6061-T6 uncoated	310	289.06	7.24	242.13	28.02	12.80	4.35	194.25	3.63	251.91
7075-T0	260	234.64	10.80	196.52	32.29	15.00	5.76	160.68	4.78	213.58
7075-T6	530	508.87	4.15	402.28	31.74	14.00	4.42	216.57	3.55	294.26

## Data Availability

The original contributions presented in the study are included in the article, further inquiries can be directed to the corresponding author.

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
