# Peer review of "Notches and Fatigue on Aircraft-Grade Aluminium Alloys"

_materials, 2024, doi:10.3390/ma17184639_

Round 1
Reviewer 1 Report
Comments and Suggestions for Authors
Dear Authors,
You have done a lot of efforts but the paper requires some corrections. Please follow the file.
Reviewer

Minor editing of English language required.
Author Response
|
Comments 1: Title needed to be changed |
|
Response 1: We have changed the title to “Notches and fatigue on aircraft-grade aluminium alloys.”
|
|
Comments 2: Due to the depth of the artificial defects and the thickness of the sample, I propose to use the name notch. |
|
Response 2: We agree with the changing the term from scratch to notch. We have change it thru the text, abbreviations, tables, figures and equation. For further clarification we have added this text: “In this experimental study, the defects are V-shaped with a considerable depth relative to the thickness of the specimens. Therefore, the term "notches" refers to these defects.” (lines 158 and 159).
|
|
Comments 3: Would you like to add a comments why you select the T4 if T6 is used very often in aircraft industry. / Would you like to add a few words why you select different strength state of the alloy examined / Scratches should be also explain with respect to operational process. |
|
Response 3: There was mistake made here, it is T3 and not T4. To answer to some of the comments, we have worked in collaboration with AEROSTAR S.A., Bacău, Romania, an aviation company specialized in manufacturing and maintenance, as indicated in the acknowledgment section. Several questions were raised by them on what would happen to the aircraft skin if notches were made by a bird beak, different object on the runway or by their workers when maintenance is performed. Therefore, they give us the materials used in this study as this is what are they currently working with. The first step of the process was to give an overall conclusion on how UTS and elongation behave under these conditions.
|
|
Comments 4: Which kind of measurement did you used to collect geometrical features of the measurement zones ? |
|
Response 4: The fallowing paragraph was added starting with line 273. We do believe that this will clarify the methodology. “The methodology used for the execution of the notches required validation to ensure control over the resulting geometry. The process precision and the wear of the scratching tool were assets by cutting from every 10th specimen a 5 mm-thick slice using an Micromet Evolution metallographic cutting machine (Remet, Bolognia, Italy). Linear and angular measurements were made on the Mitutoyo PH-3515 profile measuring projector. No significant differences were identified between the measurements and the programmed parameters (depth and length); therefore, Equations (1) thru (5) were used to generate the dimensions indicated in Table 3.”
|
|
Comments 5: Please write a few words on notch orientation versus a force direction. / Would you like to reconsider this sentence because the loading is used for the gripping section ? |
|
Response 5: The paragraph was modified and is in the manuscript starting with line 319. Also, in figure 7a the force is indicated. “All 120 specimens were subjected to the tensile test, conducted at a strain rate of 10 mm/min, with the load direction perpendicular to the transversal plane, as indicated in Figure 7a. This implies that the load is in the direction of the 0° notches and perpendicular to those oriented at 90°. The cross-section area for each specimen is indicated in Table 3 The tensile test ended when the specimen broke (Figure 7b), with the recorded data being processed for UTS and elongation at break”
|
|
Comments 6: This photo should be focused on the crack. |
|
Response 6: The image was changed, and it is focused on the crack.
|
|
Comments 7: Please add comments why you select this kind of notches its orientation covering their localisation on a one specimen side. |
|
Response 7: As indicated at response 3, the request was to analyze defects at different angles. At this stage of the research, it was not possible to analyze the notch angle with small increments, as intensive work needs to be done in this case (so far a lot of time and effort was required for this experimental plan). A much suited approach was to use 0, 45 and 90 degrees, with the mathematical models offering results for each value included in this interval. Further research will include 30 and 45 degree angles for the notch.
|
|
Comments 8: From practical point of view the uncernity follows quality of results. Would you like to add comment ? |
|
Response 8: We consider that by adding the information indicated (particularly on the measurements methodology) and by changing term such as “scratches” and “samples” we managed to better explain our results.
|
|
Comments 9: Please reduce the number of digital place to only two. In the other case please explain why you select this number of digits. |
|
Response 9: We have reduced the number of digits to only two.
|
|
Comments 10: This section is too long. Please focused you attention for fundamental points. |
|
Response 10: The conclusion section was modified as suggested.
|
|
Comments 11: Elongation at break? |
|
Response 11: We consider that maybe we did not properly explain the elongation. The values analysed are at the point of breaking, therefor this is referred as elongation at break. Maybe total elongation is a more suitable term in this case? |
Reviewer 2 Report
Comments and Suggestions for Authors
This is a very good article presenting voluminous research, with lots of data and results.
The discussion and conclusions are adequately related to conducted experiments.
However, all the conclusions are quite known, almost no new relations between the scratches influence and material properties degradation were presented.
When stating the known fact that fatigue properties of the surfaces depend on the chemical composition of the alloys, authors did not go into detailed analysis. Concretely, which elements, or their combinations, cause different behavior of these alloys?
What concerns the style of the presentation, there is a remark related to using phrases where the objects are given properties of “doing” something; like study/studies (e.g. lines 10, 44, 129, 328, etc.), results (line 19), simulation (line 144), paper (line 326). Objects cannot do anything; authors do or did. Please, reformulate all these and similar phrases using the passive voice.
The technical text should be written in the neutral form, namely in the third person, not the first. Try avoiding using “we” or “our”.
Some equations in section 3 are not “announced”, i.e. not mentioned before their appearance in the text.
Equations in the text must be written within parentheses, like “… equation (3) is …”.
The abbreviated forms of verbs (like don’t, doesn’t, etc.) are not allowed in a scientific article, since that is considered as the casual expressing.
Some abbreviations are not defined in the text. All the abbreviations must be defined/explained at the first appearance in the text.
The scanned pages of the manuscript, with marked errors and suggested corrections, are enclosed.

Comments on the Quality of English LanguagePlease, consult the scanned pages of the manuscript.
Author Response
|
Comments 1: When stating the known fact that fatigue properties of the surfaces depend on the chemical composition of the alloys, authors did not go into detailed analysis. Concretely, which elements, or their combinations, cause different behavior of these alloys? |
|
Response 1: We agree with this comment. Therefore, we have removed this paragraph as the study was directed on the mechanical properties.
|
|
Comments 2: What concerns the style of the presentation, there is a remark related to using phrases where the objects are given properties of “doing” something; like study/studies (e.g. lines 10, 44, 129, 328, etc.), results (line 19), simulation (line 144), paper (line 326). Objects cannot do anything; authors do or did. Please, reformulate all these and similar phrases using the passive voice. |
|
Response 2: We reformulate the phrases as indicated.
|
|
Comments 3: The technical text should be written in the neutral form, namely in the third person, not the first. Try avoiding using “we” or “our”. |
|
Response 3: We reformulate and avoided using “we” or “ our”.
|
|
Comments 4: Some equations in section 3 are not “announced”, i.e. not mentioned before their appearance in the text. / Equations in the text must be written within parentheses, like “… equation (3) is …”. |
|
Response 4: We have mentioned in the text the equations and also place the equation number between parentheses.
|
|
Comments 5: The abbreviated forms of verbs (like don’t, doesn’t, etc.) are not allowed in a scientific article, since that is considered as the casual expressing. |
|
Response 5: We have considered this remark and modified as suggested.
|
|
Comments 6: Some abbreviations are not defined in the text. All the abbreviations must be defined/explained at the first appearance in the text. |
|
Response 6: All abbreviations are now defined and explained first time they appear in the text.
|

Round 2
Reviewer 1 Report
Comments and Suggestions for Authors
Dear Authors,
You have done a good job. I recommend the paper for publication.
Reviewer
Comments on the Quality of English LanguageMinor editing of English language required.